# GCAF(TMEM251) regulates lysosome biogenesis by activating the mannose-6 -phosphate pathway

Weichao Zhang [1,4], Xi Yang [1,4], Yingxiang Li[1], Linchen Yu[1], Bokai Zhang[1], Jianchao Zhang [1], Woo Jung Cho[2], Varsha Venkatarangan[1], Liang Chen[1], Bala Bharathi Burugula [3], Sarah Bui[1], Yanzhuang Wang [1], Cunming Duan [1], Jacob O. Kitzman [3] & Ming Li[1] ✉

The mannose-6-phosphate (M6P) biosynthetic pathway for lysosome biogenesis has been studied for decades and is considered a well-understood topic. However, whether this pathway is regulated remains an open question. In a genome-wide CRISPR/Cas9 knockout screen, we discover TMEM251 as the first regulator of the M6P modification. Deleting TMEM251 causes mistargeting of most lysosomal enzymes due to their loss of M6P modification and accumulation of numerous undigested materials. We further demonstrate that TMEM251 localizes to the Golgi and is required for the cleavage and activity of GNPT, the enzyme that catalyzes M6P modification. In zebrafish, TMEM251 deletion leads to severe developmental defects including heart edema and skeletal dysplasia, which phenocopies Mucolipidosis Type II. Our discovery provides a mechanism for the newly discovered human disease caused by TMEM251 mutations. We name TMEM251 as GNPTAB cleavage and activity factor (GCAF) and its related disease as Mucolipidosis Type V.

The lysosome is an essential organelle responsible for the digestion and recycling of numerous cellular materials. It contains over 50 lumenal enzymes to carry out its hydrolysis function. Most enzymes utilize the mannose-6-phosphate (M6P) residues as a sorting signal for proper trafficking to the lysosome[1,2].

Because of its importance, the M6P biosynthetic pathway and its sorting mechanism have been studied extensively and considered well-understood subjects. At the cis-Golgi, GlcNAc-1-phosphate transferase (GNPT) utilizes UDP-GlcNAc as the substrate to transfer GlcNAc-1-phosphate to specific mannose residues of lysosomal enzymes. The GlcNAc molecule is then removed by an uncovering enzyme (UCE) to generate the M6P monoester. Then, lysosomal enzymes are sorted at the trans-Golgi network (TGN) by mannose-6-phosphate receptors (MPRs). MPRs traffic between the TGN, where they bind lysosomal enzymes, and endosomes, where they discharge the enzymes due to a low lumenal pH[1,3]. After releasing the substrates, MPRs are recycled back to TGN by the retromer machinery at the early endosome[4,5] or the Rab9-Tip47 machinery at the late endosome[6,7]. The discharged lumenal enzymes in endosomes are delivered to the lysosome through endomembrane trafficking.

Disruption of M6P biogenesis or its sorting results in the mistargeting of most lysosomal enzymes[8]. Mutations in GNPT cause two distinct lysosome storage diseases (LSDs)[8,9]. Mucolipidosis type II (MLII), also known as I-cell disease, is characterized by the total loss of the GNPT enzyme, whereas MLIII manifests a partial loss of enzymatic activity. MLII patients develop severe skeletal dysplasia, short stature, cardiomegaly, mental retardation, and usually die within the first decade. In contrast, MLIII patients show a later onset of similar clinical symptoms. In early 2021, a new type of LSD similar to MLII was reported[10]. However, this disease is caused by mutations in TMEM251,

[1]Department of Molecular, Cellular, and Developmental Biology, University of Michigan, Ann Arbor, MI 48109, USA. [2]BRCF Microscopy Core, University of Michigan Medical School, Ann Arbor, MI 48109, USA. [3]Department of Human Genetics, University of Michigan Medical School, Ann Arbor, MI 48109, USA. [4]These authors contributed equally: Weichao Zhang, Xi Yang. ✉e-mail: mlium@umich.edu

a gene of unknown function, indicating that our knowledge of M6P biogenesis is incomplete.

In this study, we conducted a genome-wide CRISPR/Cas9 knockout (KO) screen to identify genes essential for lysosome function. Independently, we discover TMEM251 as a critical regulator of M6P modification and lysosome biogenesis. Deleting TMEM251 causes hypersecretion of most lysosomal enzymes due to their loss of M6P and accumulation of undigested materials. To compensate for the loss of lysosome function, the cell drastically upregulates lysosome numbers (up to 5–6-fold). We further demonstrate that TMEM251 localizes to the Golgi and is required for the selective cleavage of GNPTAB by the site-1-protease (S1P), a step necessary for catalytic activation. In zebrafish, TMEM251 deletion leads to severe developmental defects in zebrafish embryos, including heart edema, insufficient cartilage, and skeletal dysplasia, which phenocopies the GNPTAB knockout. Thus, our study uncovers a novel regulator of the M6P pathway and provided the mechanism for a new human disease. We name TMEM251 as GNPTAB cleavage and activity factor (GCAF, implying that GNPT needs to be "caffeinated/activated") and its related LSD as Mucolipidosis type V.

## Results

### A genome-wide CRISPR/Cas9 screen reveals essential components for lysosome function

Previously, we and others have identified a conserved ubiquitin- and ESCRT-dependent pathway that controls the turnover of lysosome membrane proteins (LMP)[11–21]. In a recent study, we discovered two fast-degrading human LMPs, RNF152 and LAPTM4A, that are constitutively ubiquitinated and internalized into the lysosome lumen by the ESCRT machinery for degradation[17].

To identify new factors that are critical for lysosome function, we designed a genome-wide CRISPR/Cas9 KO screen in HEK293 cells. First, we generated a reporter cell line stably expressing both GFP-RNF152 and cytosolic mCherry using the internal ribosome entry site (IRES, Fig. 1a). Because GFP-RNF152 is constitutively degraded, stopping protein synthesis with cycloheximide (CHX) leads to a rapid reduction of the GFP signal. In contrast, mCherry remains stable (Fig. 1b, c). Then, we sequentially transduced the reporter line with Lenti-Cas9 and the Brunello human CRISPR library (Fig. 1d)[22]. Disruption of genes essential for lysosome function should stabilize GFP-RNF152 by blocking its degradation, leading to an elevated GFP/mCherry ratio. We performed two rounds of fluorescence-activated cell sorting (FACS) to enrich this population. After the second round, over 90% of the cell population had a stably high GFP/mCherry ratio at steady-state, and the GFP signal no longer decreased after the CHX treatment (Fig. 1e). By sequencing sgRNAs at baseline and after each round of sorting for elevated GFP/mCherry ratio, we identified 196 enriched genes from the first-round sorting and 27 genes from the second-round sorting with FDR $< 10^{-5}$ and $\log_2|FC| > 1$ (Fig. 1f, Supplementary Data 1).

Our screen results highlighted the importance of the following functional groups: (1) M6P modification at the Golgi apparatus (GNPTAB, GNPTG, and MBTPS1), (2) endosomal trafficking, regulation, and fusion machinery (HOPS & CORVET components including VPS11/16/18/33A/39/41, LRRK2, and PIKFYVE), and (3) v-ATPase components on the lysosomes and its assembly factors (ATP6V0B/C/D1, ATP6V1A/B2/C1/D/G1, ATP6AP1/2, WDR7, and VMA21) (Fig. 1f)[17]. Among the top 10 hits from the second round sorting, TMEM251, a gene of unknown function, stood out as the highest hit after RNF152, the reporter of the assay (Fig. 1g). Besides this study, TMEM251 has recently been identified by other high throughput screens designed to study autophagy, cholesterol metabolism, and lipid transport[23–25]. We decided to further characterize TMEM251 because of its implication in various biological processes and disease connection.

### TMEM251 is essential for general lysosome functions

To verify the importance of TMEM251 in LMP degradation, we knocked out the gene using two independent sgRNAs (Fig. 2a). Consistent with the screen results, both KOs significantly delayed the degradation of GFP-RNF152, and its steady-state protein level increased two-fold (Fig. 2a–c). Besides RNF152, we also tested another fast-degrading LMP, LAPTM4A[17]. As shown in Fig. 2d–f, TMEM251 deficiency led to a drastic increase in the endogenous LAPTM4A protein level (8–10-fold) and much slower degradation kinetics.

We then asked if TMEM251 is also involved in other lysosome-dependent processes, such as the degradation of cell surface receptors via endocytosis. Since the expression level of epidermal growth factor receptor (EGFR) in HEK293 cells is low[26,27], we generated two independent HeLa KO cell lines to evaluate its degradation induced by EGF[28]. Our data showed that TMEM251 KO leads to a significant delay in EGFR turnover (Fig. 2g, h). Besides endocytosis, TMEM251 KO cells also have defects in autophagy. As shown in Fig. 2i–k, TMEM251 deficiency led to a two-fold increase in p62/SQSTM1 protein level and a 6-fold increase in the lipidated LC3B-II level under normal growth conditions.

In human cells, *TMEM251* encodes two transcriptional variants through alternative splicing: a long isoform (18.7 kDa) and a short isoform (15.2 kDa) (Supplementary Fig. 1a). Using an antibody that recognizes the C-terminus of TMEM251, we verified that the short isoform is the predominant variant (Supplementary Fig. 1b). Importantly, both long and short isoforms can rescue the degradation of LAPTM4A and LC3-II, indicating that they are functionally redundant in lysosome biogenesis (Supplementary Fig. 1c–e). For the rest of the study, we focused on the short isoform unless mentioned otherwise.

As TMEM251 deficiency impairs various lysosome-dependent pathways, including LMP degradation, EGFR endocytosis, and autophagy, we concluded that TMEM251 is a critical factor in regulating general lysosome function.

### Ablation of TMEM251 upregulates lysosome biogenesis

How does TMEM251 deficiency lead to a general lysosome defect? We envisioned that two scenarios might explain this observation: (1) lysosomes might have an acidification defect, which inactivates lumenal hydrolases that depend on the low pH to be functional, or (2) the lumenal hydrolases might be absent from lysosomes. To test the first hypothesis, we stained cells with LysoTracker Red DND-99 that labeled the acidic endo-lysosomes and analyzed them by flow cytometry. Instead of lowering the fluorescent intensity, TMEM251 KO led to a drastic increase in the lysotracker signal in both HeLa and HEK293 cells (4–6-fold, Fig. 3a, b, Supplementary Fig. 2), indicating that the v-ATPase function is not impaired. Using transmission electron microscopy, we observed a massive increase of electron-dense lysosomes with undigested materials accumulating inside (Fig. 3c, d). The average lysosome radius also increases by 10%, corresponding to a ~30% increase in volume (Fig. 3d, e).

Intrigued by the increase in lysosome numbers, we analyzed the KO cells with RNA sequencing (Supplementary Fig. 3a). Transcriptome analysis reveals 211 differentially expressed genes (DEGs) with $p < 0.05$ and $\log_2|FC| > 0.263$ (Supplementary Fig. 3b). The gene ontology (GO) analysis confirmed the upregulation of lysosome pathways at the transcriptional level (Fig. 3f–h). When categorizing these DEGs into biological processes, we noticed that genes involved in lipid metabolism, autophagy, and UDP-GlcNAc biosynthesis were also upregulated (Supplementary Fig. 3c, d). These findings are consistent with the recent CRISPR/Cas9 KO screens suggesting that TMEM251 KO leads to increased lipid/cholesterol biosynthesis[23,25]. As the lysosome plays a critical role in regulating lipid/cholesterol homeostasis, the upregulation may be due to the feedback responses for lysosome dysfunction.

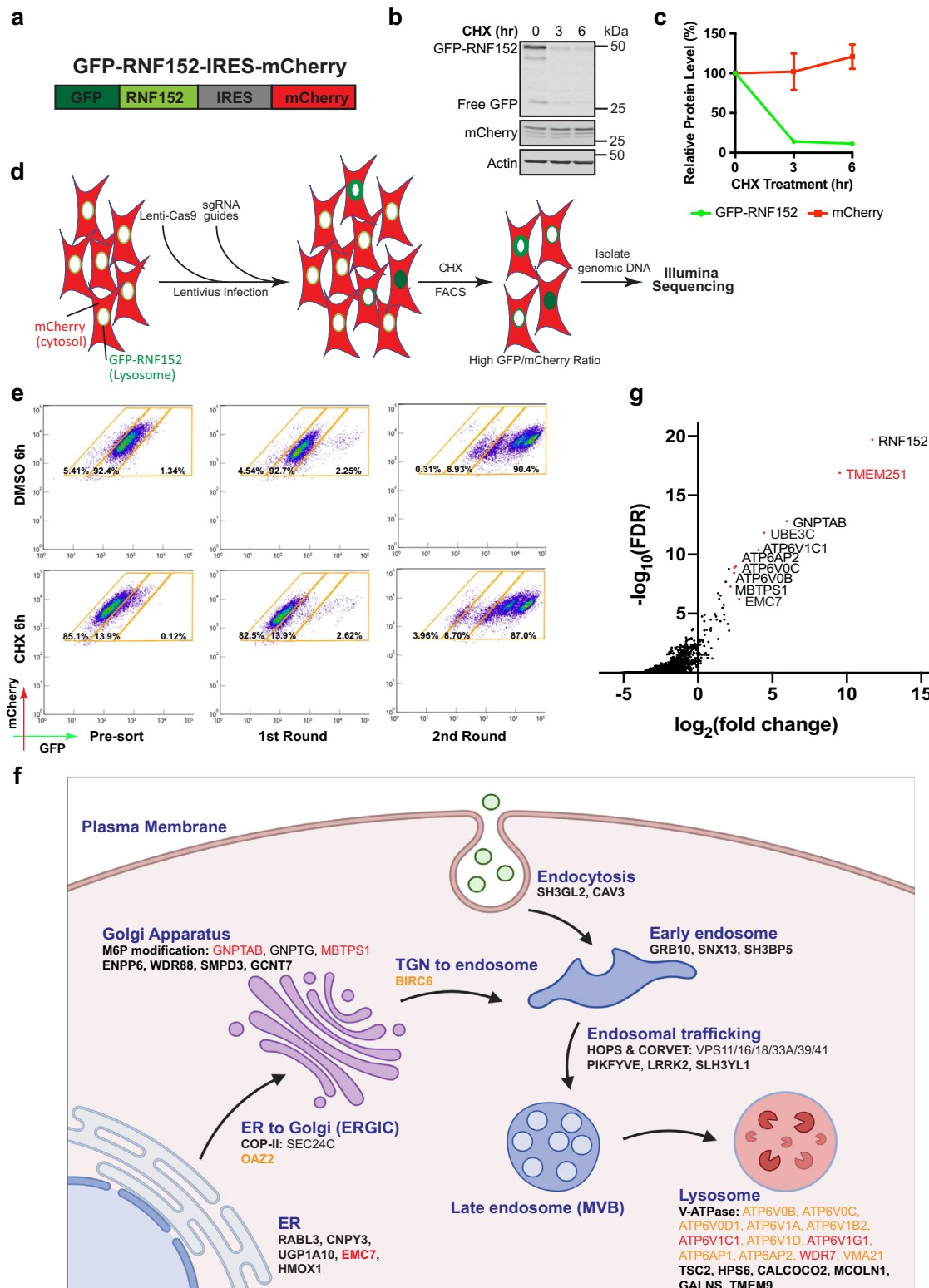

**Fig. 1 | A genome-wide CRISPR-Cas9 knockout screen to identify genes essential for lysosome function. a** A schematic representation of the reporter construct. **b** CHX chase assay of stably expressed GFP-RNF152 and mCherry in HEK293 cells. **c** Quantification of the protein levels in **b**. Mean of 3 independent replicates is shown. Error bars represent standard deviation. **d** A schematic representation of the CRISPR-Cas9 screen to identify genes essential for lysosome function. **e** Flow cytometry profiles of presorted and sorted cells from the CRISPR-Cas9 screen with or without CHX treatment. **f** A schematic representation of hits from the CRISPR-Cas9 screen that highlights the membrane trafficking pathways (Created with BioRender.com). Black: hits from the first-round sorting. Orange: hits from the second-round sorting. Red: hits appeared in both rounds of sorting. **g** Top 10 hits of Illumina sequencing of sgRNAs from the second round of sorting.

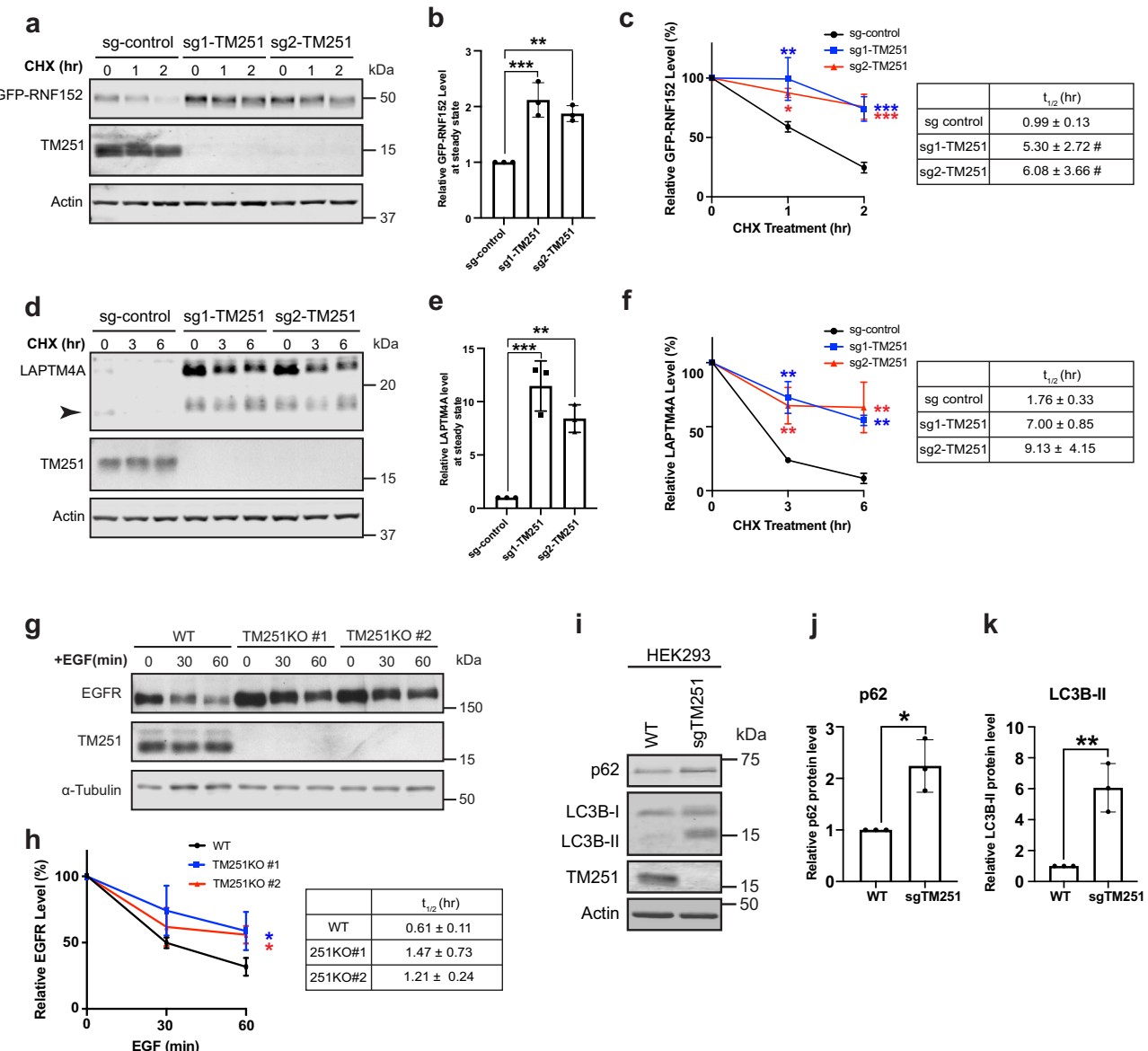

**Fig. 2 | TMEM251 deficiency leads to lysosome dysfunction. a** CHX chase assay of stably expressed GFP-RNF152 in sgRNA control, sgRNA-1 TMEM251, and sgRNA-2 TMEM251 cells. **b** Steady-state (0 h) protein levels in **a**. Mean of 3 independent replicates is shown. Error bars represent standard deviation. **$p \leq 0.01$, ***$p \leq 0.001$. **c** Quantification of GFP-RNF152 degradation in **a**. Mean of 3 independent replicates is shown. Error bars represent standard deviation. *$p \leq 0.05$, **$p \leq 0.01$, ***$p \leq 0.001$. Table: Calculated protein half-lives. **d** CHX chase assay of endogenous LAPTM4A in sgRNA control, sgRNA-1 TMEM251, and sgRNA-2 TMEM251 cells. Arrowhead: cleavage product of LAPTM4A. **e** Steady-state (0 h) LAPTM4A protein levels in **d**. Mean of 3 independent replicates is shown. Error bars represent standard deviation.

**$p \leq 0.01$, ***$p \leq 0.001$. **f** Quantification of LAPTM4A degradation in **d**, Mean of 3 independent replicates is shown. Error bars represent standard deviation. **$p \leq 0.01$. Table: Calculated protein half-lives. **g** EGFR degradation assay in HeLa WT and TMEM251 KO cells. **h** Quantification of EGFR degradation in **g**. Mean of 3 independent replicates is shown. Error bars represent standard deviation. *$p \leq 0.05$. Table: Calculated protein half-lives. **i** p62 and LC3B protein levels in HEK293 WT and sgTMEM251 cells. **j**, **k** Quantification of the p62 (**j**) and LC3B-II (**k**) protein levels in (**i**). Mean of 3 independent replicates is shown. Error bars represent standard deviation. *$p \leq 0.05$. **$p \leq 0.01$. See source data file for exact P values.

## TMEM251 deficiency leads to the secretion of many lysosomal hydrolases

After ruling out the acidification defect, we tested if the lumenal hydrolases can properly target to the lysosome using Cathepsin C and D (CTSC and CTSD) as proxies. Both enzymes are sorted at TGN by M6PR and undergo sequential cleavage into mature forms when trafficking to the lysosome. Strikingly, TMEM251 deletion abolished the mature forms of both enzymes, and only the accumulated proenzymes were detected (Fig. 4a–c). Furthermore, analyzing the conditioned culture media indicated that a significant portion of ProCTSC and ProCTSD were secreted into the media (Fig. 4a–c) in TMEM251 deficiency cells.

To obtain a holistic view of how many lysosomal enzymes are secreted, we compared the conditioned media from TMEM251 KO to WT cells using quantitative mass spectrometry. Our analysis uncovered 39 lumenal proteins exhibiting a significant increase in secretion ($\log_2|$FC$| > 1$ and $p < 0.05$) (Fig. 4d, e, Supplementary Table 1), indicating that most lumenal proteins are mistargeted in TMEM251 KO cells. This result explained why lysosomes are defective after deleting TMEM251.

## TMEM251 is essential for the M6P biogenesis of lysosomal enzymes

How does TMEM251 deficiency lead to the hypersecretion of lysosomal enzymes? We first determined its subcellular localization. Using its

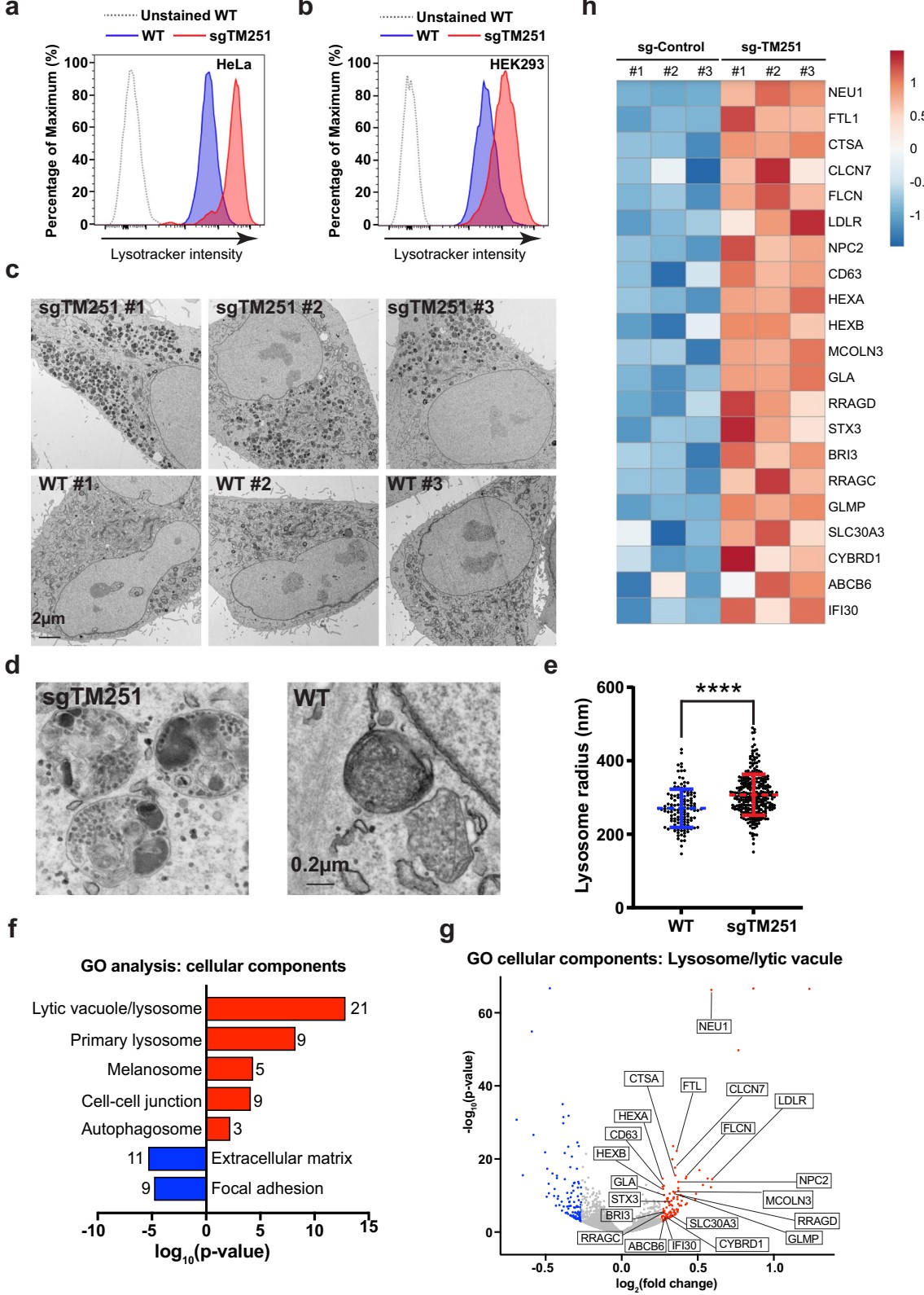

**Fig. 3 | Ablation of TMEM251 upregulates lysosome biogenesis. a**, **b** Lysotracker intensity of WT vs. sgTMEM251 in HeLa (**a**) and HEK293 (**b**) cells. **c** Representative TEM images of HeLa WT (*n* = 100) and sgTMEM251 (*n* = 100) cells. Scale bar: 2 μm. **d** Representative zoomed-in TEM images of lysosomes in HeLa WT (*n* = 100) and sgTMEM251 (*n* = 100) cells. Scale bar: 0.2 μm. **e** Quantification of lysosome radius in HeLa WT (*n* = 131) and sgTMEM251 (*n* = 297) cells. Mean of the quantification is shown with error bars representing the standard deviation. ****$p \leq 0.0001$. See

source data file for exact *P* values. **f** Metascape Gene Ontology (GO) cellular component analysis of differentially expressed genes (DEGs) altered in sgTMEM251 (*n* = 3 independent replicates) vs. control (*n* = 3 independent replicates) cells. The number of genes in each pathway is indicated. **g** Volcano plot of RNA-seq analysis of sgTMEM251 (*n* = 3 independent replicates) vs. control (*n* = 3 independent replicates) cells. Annotated genes are classified in lysosome/lytic vacuole components. **h** Heatmap of genes annotated in **g**.

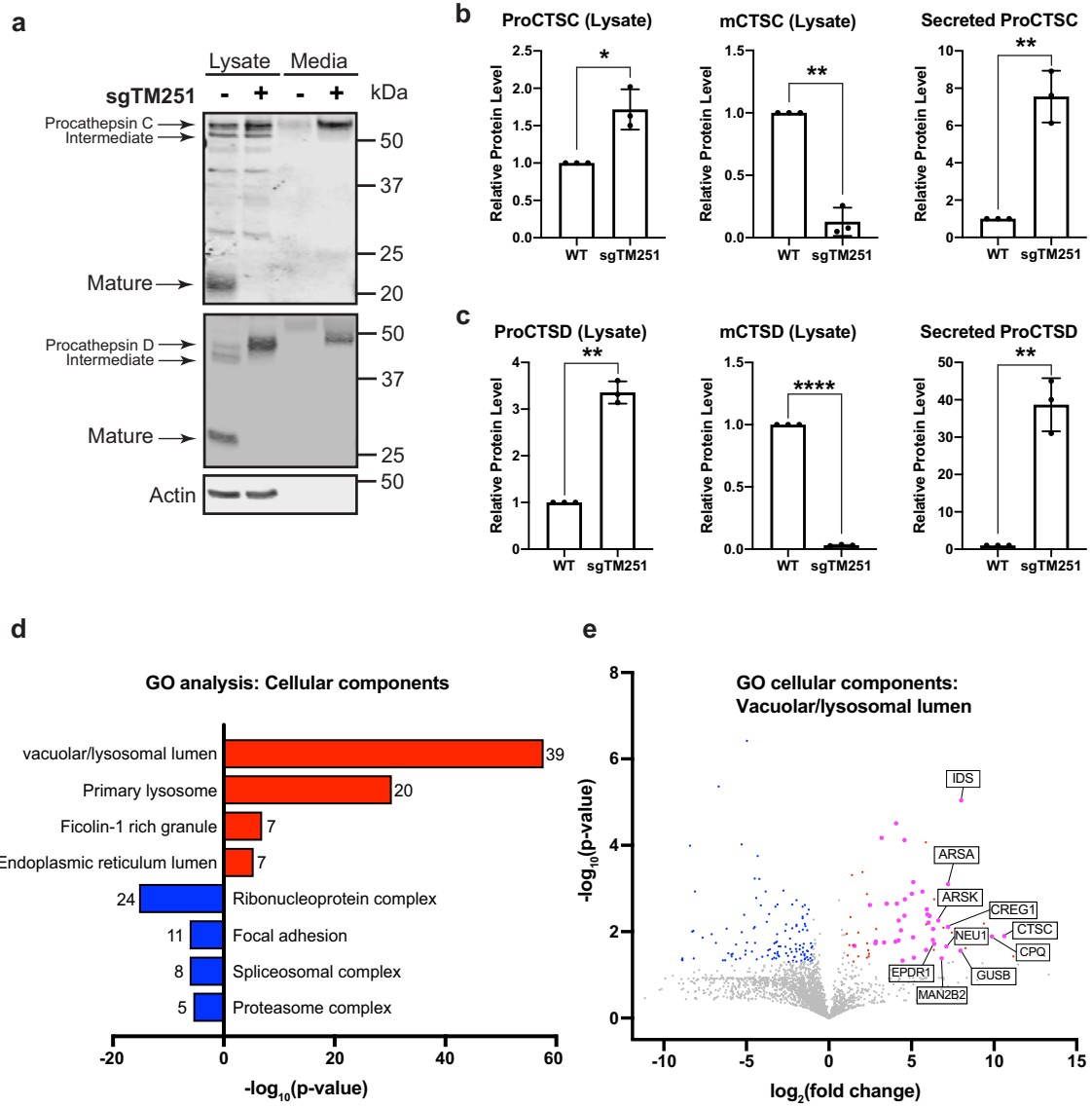

**Fig. 4 | TMEM251 deficiency leads to hypersecretion of lysosomal enzymes.**
**a** CTSC and CTSD protein level in the whole cell lysate and conditioned media of HEK293 WT and sgTMEM251 cells. **b**, **c** Quantification of CTSC and CTSD protein levels in **a**. Mean of 3 independent replicates is shown. Error bars represent standard deviation. *$p \leq 0.05$, **$p \leq 0.01$, ****$p \leq 0.0001$. See source data file for exact $P$ values. **d** GO enrichment analysis of secreted proteins altered in sgTMEM251 ($n = 3$ independent replicates) vs. control ($n = 3$ independent replicates) cells. The number of proteins in each pathway is indicated. **e** Volcano plot of secretome analysis of sgTMEM251 ($n = 3$ independent replicates) vs. control ($n = 3$ independent replicates) cells. Annotated genes are the top 10 candidates in vacuolar/lysosomal lumenal components.

endogenous antibody, we observed that both long and short isoforms of TMEM251 colocalized with the cis-Golgi marker GM130 (Supplementary Fig. 4a, b), which is consistent with the previous report that TMEM251-GFP is localized to Golgi[10]. A small fraction of TMEM251 also colocalized with the early endosome marker EEA1 and the lysosome marker LAMP2 (Supplementary Fig. 4a, b). Consistent results were obtained using biochemically purified organelles. As shown in Supplementary Fig. 4c, d, endogenous TMEM251 is highly enriched in purified rat liver Golgi membranes and slightly enriched in immuno-isolated lysosomes.

Besides TMEM251, our CRISPR screen also identified three other Golgi factors, including *GNPTAB*, *GNPTG*, and *MBTPS1* (Fig. 1f, g). *MBTPS1* encodes site-1 protease (S1P), a Golgi-resident serine protease that is responsible for the proteolytic cleavage of multiple substrates, including GNPTAB, SREBP1/2, and ATF6[29,30]. *GNPTAB* and *GNPTG* encode the three subunits of the GNPT enzyme[31], which requires S1P cleavage for its activation and is responsible for adding M6P to the

lysosomal enzymes (Fig. 5a). Then the lysosome enzymes are sorted at TGN by two types of M6P receptors (MPRs), the ~46-kDa cation-dependent MPR (CD-MPR) and the ~300-kDa cation-independent MPR (CI-MPR), for their lysosomal delivery[1,3]. Therefore, mutations in either GNPT, S1P, or MPRs will result in M6P biogenesis/sorting defects and the secretion of most lysosomal enzymes. Interestingly, published bioinformatic analysis indicated a co-dependency (Pearson correlation 0.26, https://www.depmap.org) between TMEM251 and GNPTAB from large-scale CRISPR KO screen datasets, suggesting a genetic correlation between the two genes. Because of similar localization, consistent enzyme secretion phenotypes, and genetic correlation, we hypothesized that TMEM251 is a critical factor in the M6P biogenesis pathway.

To test the hypothesis, we compared the processing and secretion of proCTSD in TMEM251 KO, GNPTAB KO, and CI-MPR KO cells (Fig. 5a, b, Supplementary Fig. 5a). TMEM251 KO phenocopied GNPTAB deficiency as evidenced by the absence of mCTSD and the accumulation of ProCTSD[32]. CI-MPR KO cells showed a mild phenotype with a small

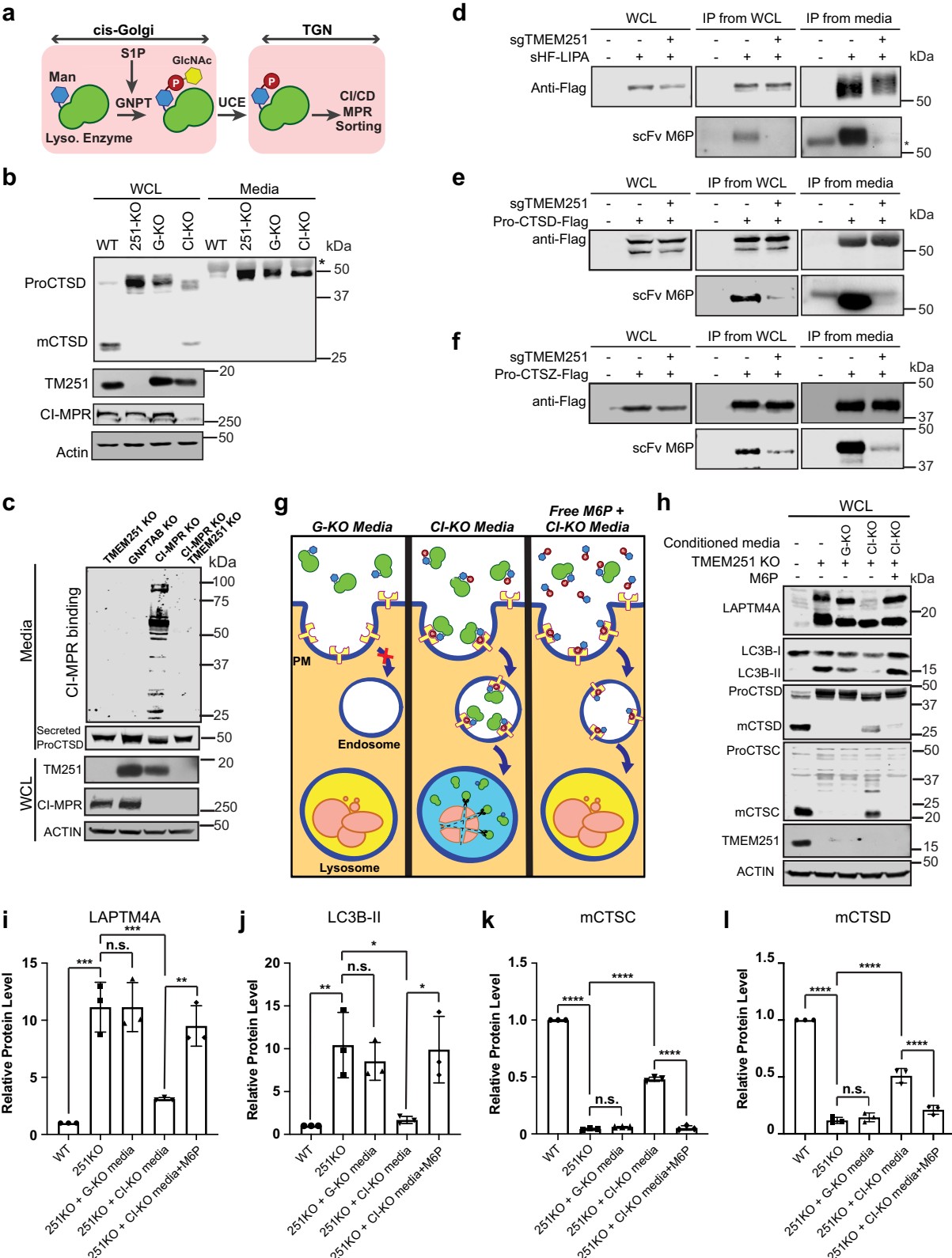

**Fig. 5 | TMEM251 deficiency leads to M6P modification defects of lysosomal enzymes. a** A schematic representation of M6P modification and sorting of lysosomal enzymes. **b** CTSD protein level in the whole cell lysate and conditioned media of HEK293T WT, TMEM251 KO (251-KO), GNPTAB KO (G-KO), CI-MPR KO (CI-KO) cells (*n* = 3 independent replicates). Asterisk: a non-specific band. **c** CI-MPR binding assay of conditioned media from TMEM251 KO, GNPTAB KO, CI-MPR KO, and CI-MPR and TMEM251 double KO cells (*n* = 3 independent replicates). **d**–**f** Detection of M6P modification of LIPA (**d**), CTSD (**e**), and CTSZ (**f**) in HEK293T

and sgTMEM251 cells (*n* = 2 independent replicates) using single-chain antibodies against M6P (scFv M6P). Asterisk: a non-specific band. **g**, **h** Rescue of TMEM251 KO with conditioned media from GNPTAB KO and CI-MPR KO cells. **i**–**l** Quantification of the full-length LAPTM4A, LC3B-II, mature CTSC, and mature CTSD protein levels in **h**. Mean of 3 independent replicates is shown. Error bars represent standard deviation. \**p* ≤ 0.05, \*\**p* ≤ 0.01, \*\*\**p* ≤ 0.001, \*\*\*\**p* ≤ 0.0001. See source data file for exact *P* values.

amount of proCTSD processed to the mature form, likely due to the presence of CD-MPR (Fig. 5b). Importantly, all three KO lines exhibited strong secretion of ProCTSD (Fig. 5b). To directly evaluate the M6P modification state of secreted enzymes, we used purified CI-MPR to detect the presence of M6P in the conditioned media[33,34]. Our results indicated that only the conditioned media from CI-MPR KO cells contained M6P-tagged glycoproteins (Fig. 5c). Furthermore, knocking out TMEM251 in CI-MPR KO cells abolished the M6P modification, suggesting that TMEM251 functions upstream of the M6P sorting step and is likely involved in the M6P modification (Fig. 5c). We also verified our results by examining individual enzymes, including Lipase A (LIPA), CTSD, and cathepsin Z (CTSZ). After immunoprecipitation, we used a single-chain antibody against M6P to detect the modification[35,36]. As shown in Fig. 5d–f, TMEM251 deficiency abolished the M6P modification of all three enzymes.

Next, we took advantage of the fact that ~10% CI-MPR localizes to the cell surface to endocytose secreted lysosomal enzymes (Fig. 5g)[3], and tested if enzyme replacement therapy (ERT) can be used to treat TMEM251 deficiency[37,38]. To this end, we collected conditioned media from CI-MPR KO cells that contain M6P-tagged lysosomal enzymes to feed the TMEM251 KO cells[37,39]. Indeed, conditioned media from CI-MPR KO, but not GNPTAB KO cells, partially rescued lysosome defects in TMEM251 KO cells, as evidenced by the reduction of full-length LAPTM4A and LC3B-II (Fig. 5h–j). We also observed a significant increase of mature enzymes such as mCTSC and mCTSD in cells (Fig. 5h, k, l), indicating that endocytosed enzymes have reached the lysosome. These rescue phenotypes were abolished by adding excessive free M6P, which saturated the surface M6P receptors (Fig. 5g–i). Collectively, these results demonstrated that ERT could be used to treat TMEM251 deficiency. They also confirmed that the lysosome defects in TMEM251 KO cells are due to the lack of M6P on its lumenal enzymes.

### TMEM251 is required for the efficient processing of the GNPT α/β precursor

The complete GNPT enzyme is an $\alpha_2\beta_2\gamma_2$ hexamer. It is assembled at the endoplasmic reticulum (ER) before its trafficking to the Golgi. Upon arrival at the Golgi, GNPT is cleaved between the α/β precursor by S1P and activated (Fig. 6a)[30,40]. Since our data so far suggested that TMEM251 is also a critical factor for M6P biogenesis, we decided to resolve the relationship between TMEM251, GNPT, and S1P. Due to the lack of endogenous antibodies, we used CRISPR/Cas9 technique to knock in a 3xHA tag at the C-terminus of the GNPTAB in HEK293T cells (Supplementary Fig. 5b). After knocking in, the levels of CTSD, LAPTM4A, and LC3B-II were largely unchanged, suggesting that HA tagging did not affect the GNPTAB function (Supplementary Fig. 5c). At the endogenous level, most GNPT α/β precursor is processed into the active form, as indicated by the strong signal of the β subunit at ~48 kDa. The processed β subunit was abolished in TMEM251 KO cells, and was further rescued by reintroducing TMEM251, ruling out the off-target effect of the CRISPR technique (Fig. 6a–c). These results indicated that TMEM251 is required for the cleavage and activity of GNPT by S1P, which explained why knocking out TMEM251 abolished the M6P modification. Interestingly, overexpression of TMEM251 can slightly stimulate the processing of endogenous GNPTAB (Fig. 6b, compare lanes 2 to 4). This stimulation became evident when GNPTAB was overexpressed (Supplementary Fig. 5d, e).

We then asked if the GNPT processing defect was due to a deficiency in S1P by testing other substrates of S1P. As shown in Fig. 6d–i, TMEM251 KO did not affect the processing of SREBP2 triggered by sterol depletion (Fig. 6d–f)[41,42], nor did it affect the cleavage of ATF6 triggered by DTT treatment that induces ER stress (Fig. 6g–i)[43], suggesting that only the GNPTAB cleavage is affected by TMEM251 deletion. Furthermore, GNPTAB largely colocalized with cis-Golgi Glycoprotein GPP130 in both WT and TMEM251 KO

cells (Supplementary Fig. 5f, g), suggesting TMEM251 is not required for its ER exit.

We further assessed the physical interactions among the three proteins. As shown in Fig. 6j–m, reciprocal IP experiments demonstrated that TMEM251 interacts with both GNPTAB and S1P, supporting its role in facilitating GNPTAB cleavage by S1P.

Altogether, we concluded that TMEM251 is a specific factor required for GNPTAB processing. Thus, we named TMEM251 the GNPTAB cleavage and activity factor (GCAF).

### GCAF deficiency phenocopies MLII in vivo

Bioinformatic analysis indicated that GCAF is highly conserved within vertebrate animals but not detected outside the vertebrates. The amino acid sequence of zebrafish (*Danio rerio*) GCAF shares a 69.7% identity with human GCAF (Fig. 7a) even though the two species are separated by 450 million years[44]. Importantly, overexpression of *Dr*GCAF rescued the KO phenotypes in human cells (Fig. 7b, c), indicating an evolutionarily conserved role in M6P biogenesis.

Previous studies have successfully established zebrafish models to study LSDs, including mucolipidosis type II[45–48]. Since our data indicated that GCAF is required for the cleavage and activation of GNPT α/β precursor, we asked if their knockouts share similar developmental defects in zebrafish. To this end, we knocked out GNPTAB and GCAF using the newly developed F0 knockout technique[49]. In this method, a mixture of 4 sgRNAs targeting different locations of the *GCAF* (or *GNPTAB*) gene, along with the Cas9 mRNA, was injected into fertilized eggs at the one-cell stage. The sgRNAs will convert many injected embryos into biallelic knockouts and allow us to examine developmental defects.

Both mutants displayed similar defects, with over 50% of the population showing severe edema (enlarged heart and other internal organs) and skeletal dysplasia at 5–7 dpf (day post-fertilization, Fig. 7d–f). We further categorized these defects into three groups: (1) severe edema only ($22.9 \pm 0.8\%$ for GCAF and $20.4 \pm 5.7\%$ for GNPTAB), (2) severe edema with curly tail or no tail ($29.5 \pm 11.0\%$ for GCAF and $25.5 \pm 8.7\%$ for GNPTAB), (3) curly tail or no tail only ($4.8 \pm 1.0\%$ for GCAF and $3.7 \pm 1.9\%$ for GNPTAB, Fig. 7d–f). We also examined cartilage and calcified bone structures using Alcian blue and Alizarin red staining[50,51]. As shown in Fig. 7g, h, both knockout embryos lost the majority of the cartilage and calcified structures such as bones and ear stones. These defects were consistent with the clinical observations that ML-II was commonly associated with cardiovascular abnormalities and defects in skeletal development[8,52–54].

Altogether, our in vivo study indicated that GCAF deletion phenocopies the GNPTAB KO during early embryo development, strongly supporting that both proteins function in the M6P biogenesis pathway.

## Discussion

In this study, we performed a genome-wide CRISPR KO screen to identify new factors essential for lysosome function. We discovered TMEM251/GCAF as a critical protein for lysosome biogenesis by regulating the cleavage of GNPT α/β precursor. GCAF physically interacts with both GNPTAB and S1P to facilitate protein cleavage. Ablation of GCAF results in M6P modification defects. Consequently, most lumenal enzymes cannot be sorted appropriately, leading to severe lysosome defects (Fig. 8). In zebrafish, knocking out GCAF led to severe cardiac and skeletal abnormalities that resembled MLII phenotypes. Excitingly, in cultured cells, the lysosome defects can be rescued by conditioned media containing normal M6P-labeled enzymes, which could form the basis for developing ERT to treat patients.

Mucolipidosis is classified into four subtypes according to the genes/enzymes that are affected. ML-I (sialidosis) results from the deficiency of lysosomal sialidase, which removes sialic acid from glycoproteins, leading to the accumulation of toxic carbohydrates in the cell[55]. MLII and MLIII are both associated with GNPT. MLII is

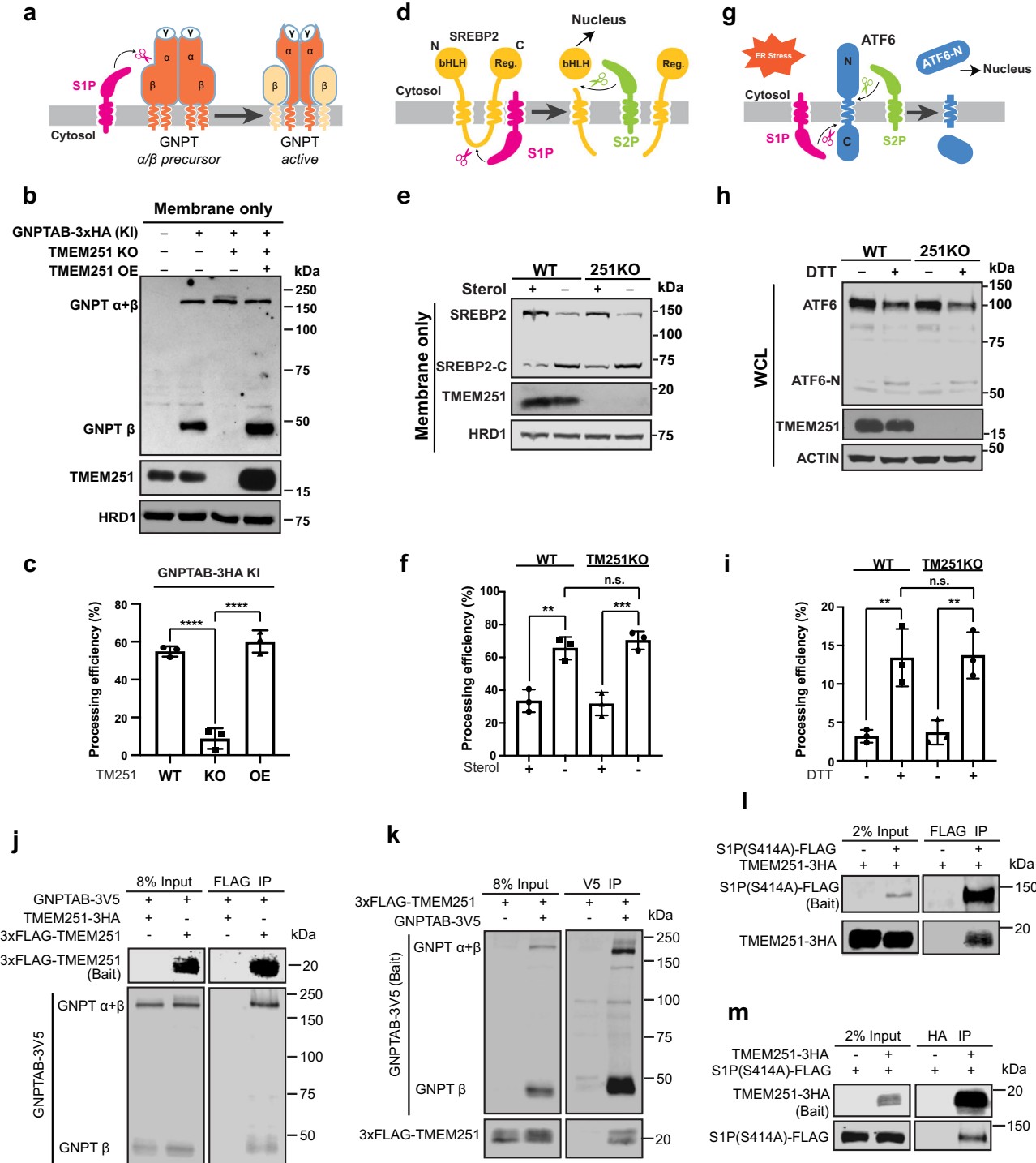

**Fig. 6 | TMEM251 is required for the efficient processing of GNPT α/β precursor by S1P. a** A schematic representation of GNPTAB processing by S1P. **b** The processing of the endogenously tagged GNPTAB in TMEM251 KO and TMEM251 overexpression (OE) cells. **c** Quantification of the GNPTAB processing efficiency in **b**. Mean of 3 independent replicates is shown. Error bars represent standard deviation. ****$p \leq 0.0001$. **d** A schematic representation of SREBP2 processing by S1P and S2P. **e** The processing of SREBP2 in HEK293T WT and TMEM251 KO cells. **f** Quantification of the SREBP2 processing efficiency in **e**. Mean of 3 independent replicates is shown. Error bars represent standard deviation. **$p \leq 0.01$, ***$p \leq 0.001$. **g** A schematic representation of ATF6 processing by S1P and S2P. **h** ATF6 processing in HEK293T WT and TMEM251 KO cells after 1 h of CHX and DTT treatment. **i** Quantification of the ATF6 processing efficiency in (**h**). Mean of 3 independent replicates is shown. Error bars represent standard deviation. **$p \leq 0.01$. See source data file for exact *P* values. **j**, **k** Reciprocal IP ($n = 2$ independent replicates) showing interactions between GNPTAB and TMEM251. **l**, **m** Reciprocal IP ($n = 2$ independent replicates) showing interactions between S1P (S414A) and TMEM251.

characterized by the build-up of waste products called inclusion bodies, and patients often die in early childhood due to heart failure or respiratory tract infection[52,56,57]. In contrast, MLIII manifests less severe symptoms and progresses slower[58]. Last, pathogenic mutations of

MCOLN1 cause ML-IV, which is characterized by delayed psychomotor development and progressive visual impairment[59]. MCOLN1 encodes a cation channel that releases $Ca^{2+}$ from endolysosomal compartments, which regulates lysosome-related events such as fusion, positioning,

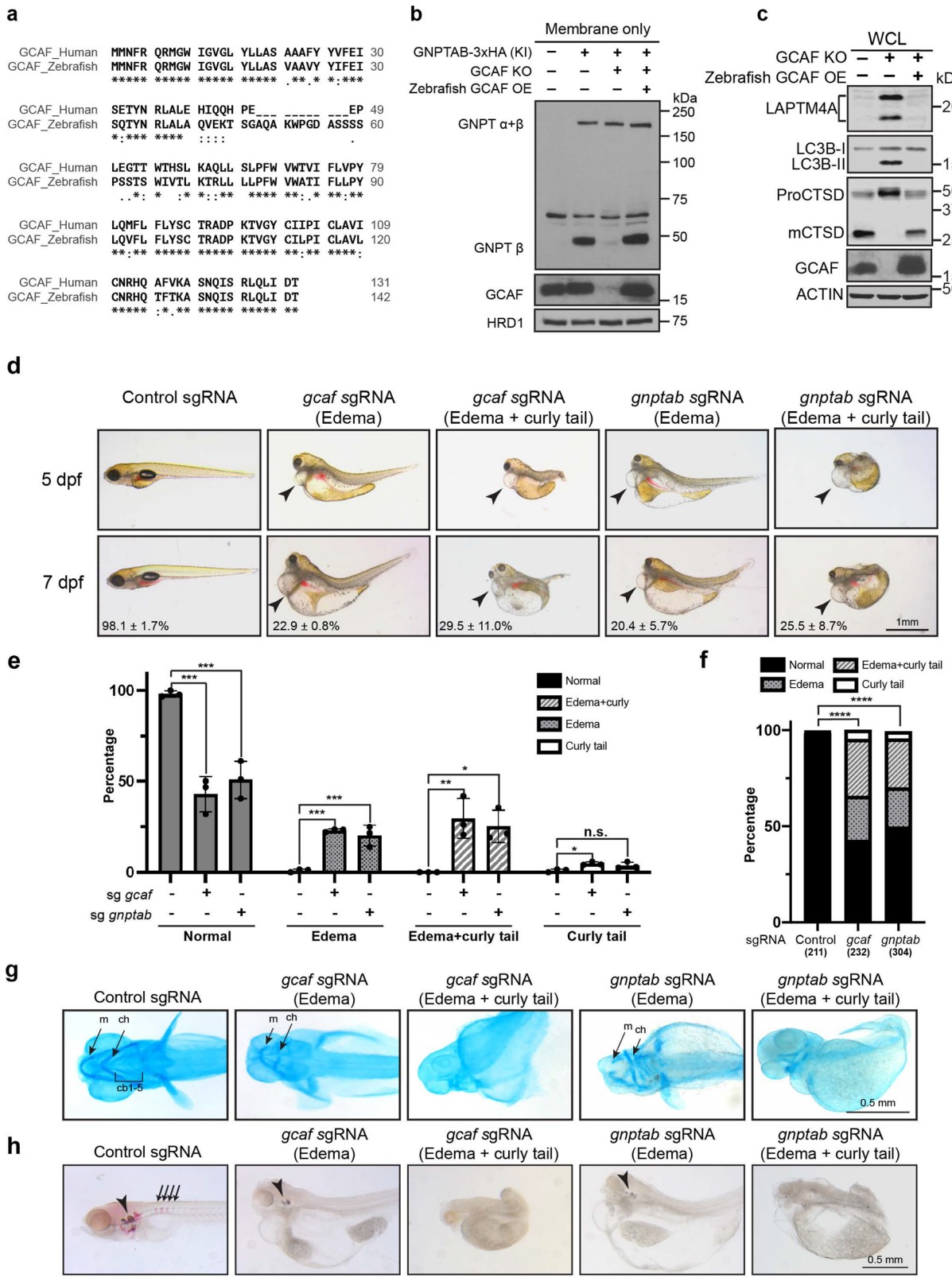

**Fig. 7 | GCAF deficiency phenocopies ML-II in vivo. a** Sequence alignments of human GCAF short isoform and zebrafish GCAF. **b** Zebrafish GCAF rescued the GNPTAB processing defects in GCAF KO HEK293T cells (*n* = 3 independent replicates). **c** Zebrafish GCAF rescued the lysosome function in GCAF KO HEK293T cells (*n* = 3 independent replicates). **d** Morphology of the F0 GCAF and GNPTAB deficient fish at 5 and 7 dpf. Arrowheads point to heart edema. **e** Quantification of morphological phenotypes observed in **d**. Mean of 3 independent replicates is shown. Error bars represent standard deviation. *\*p ≤ 0.05, \*\*p ≤ 0.01, \*\*\*p ≤ 0.001*. See source data file for exact *P* values. **f** χ² test to compare control, sgGCAF, and sgGNPTAB embryos. *\*\*\*\*p ≤ 0.0001*. The numbers in x-axis represent the fishes included in the quantification. **g** Ventral view of alcian blue stained zebrafish larvae at 4 dpf. ch: ceratohyal, m: Meckel's cartilage; cb: ceratobranchials. **h** Alizarin red staining of zebrafish embryos at 7 dpf. Arrowheads: ear stones. Arrows: vertebrate columns.

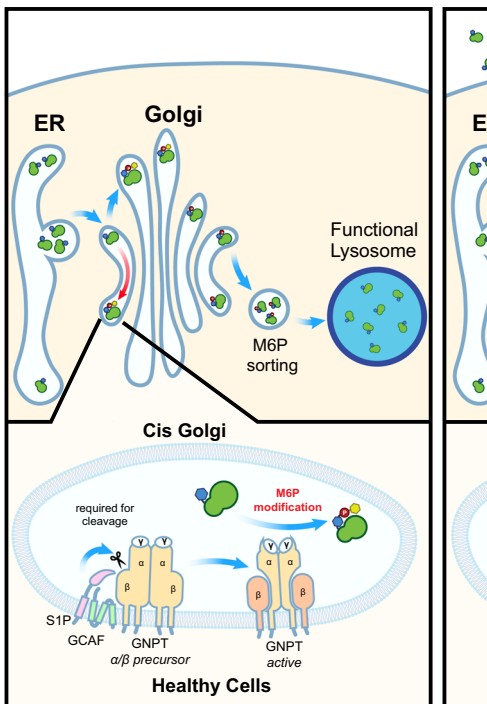
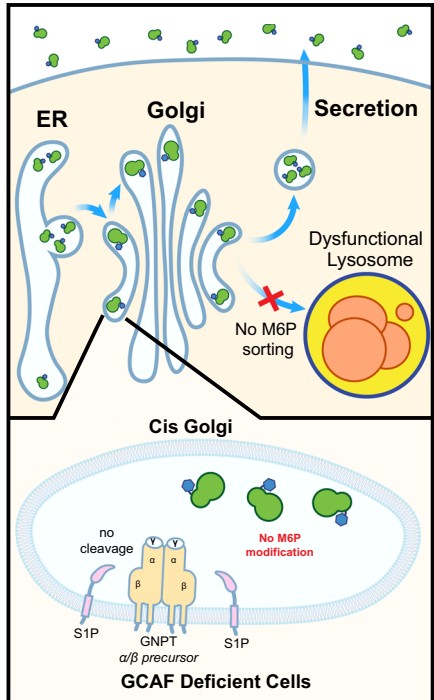

**Fig. 8 | A model showing TMEM251/GCAF is required for the cleavage and activation of GNPT α/β precursor.** TMEM251/GCAF deficiency leads to defects in M6P modification of lysosomal enzymes at the cis-Golgi. Lysosomal enzymes without M6P are targeted to the secretory pathway, resulting in lysosome dysfunction.

and lysosome exocytosis[60–63]. This study characterized a new type of LSD associated with GCAF. Mechanistically, GCAF is essential for the processing and activity of GNPT. Symptomatically, individuals who carried the pathogenic GCAF mutation displayed severe symptoms reminiscent of MLII[10]. Therefore, we propose classifying this MLII-like inherited metabolic disorder as Mucolipidosis type V.

Mounting evidence has indicated that S1P can cleave multiple diverse Golgi substrates, including SREBP1/2, GNPTAB, activating transcription factor 6 (ATF6), the secreted kinase Fam20C, and many viral glycoproteins[30,64,65]. Therefore, it is important to regulate S1P activities on different substrates. One mechanism is to modulate S1P activity directly. Recently, several independent studies have uncovered a new factor, C12orf49, that positively regulates SREBPs signaling by promoting S1P proteolytic activities[66–69]. C12orf49 is required for the maturation of S1P[69]. However, it does not provide substrate specificity[67,69].

Another elegant mechanism is modulating substrates' subcellular localization in response to specific environmental cues. For example, the localization and processing of SREBP1/2 are regulated by cholesterol levels. When cholesterol is abundant in the ER, the SREBPs/SCAP complex interacts with INSIG1/2, causing ER retention[70]. Cholesterol depletion leads to dissociation of SREBPs/SCAP from INSIGs and transport to the Golgi via COP-II vesicles[71]. At Golgi, SREBPs are sequentially cleaved by S1P and S2P, releasing the N-terminal bHLH transcription factor domain, which enters the nucleus to activate the cholesterol and lipid biosynthesis pathways[72,73].

Our study unveiled a new mechanism that regulates S1P selectivity. GCAF acts as an adapter for GNPTAB, which breaks the long-standing view that GNPTAB is constitutively cleaved and activated by S1P. It localizes to Golgi and physically interacts with both GNPTAB and S1P to modulate the cleavage. Deletion of GCAF abolished GNPTAB processing, and most lysosomal enzymes lose the M6P modification. To date, it is unclear how GCAF engages S1P and GNPTAB to activate cleavage. A careful study to map the interacting domains among the three partners would be an important future direction.

## Methods

### Ethics statement
All experiments were conducted in accordance with the guidelines approved by the Institutional Committee on the Use and Care of Animals, University of Michigan.

### Zebrafish husbandry
Zebrafish were raised following standard zebrafish husbandry guidelines[74]. Embryos were obtained by natural crosses and raised in a standard E3 embryo medium[74]. Embryos were staged as described previously[75]. To inhibit pigmentation, 0.003% (w/v) N-phenylthiourea (PTU) was added. All experiments were conducted in accordance with the guidelines approved by the University of Michigan Institutional Committee on the Use and Care of Animals.

### Preparation of Golgi membranes from rat liver
Golgi membranes were prepared from fresh liver tissues of female Sprague–Dawley rats as described previously[76]. Briefly, rats were euthanized by carbon dioxide ($CO_2$) inhalation followed by cervical dislocation after 24-h food starvation. Liver tissues were rapidly washed in PBS and transferred into ice-cold buffer C (0.5 M sucrose, 5 mM $MgCl_2$, 0.1 M phosphate buffer pH 6.7) with EDTA-free protease inhibitors and pepstatin A. Liver tissues were cut by a pair of surgical scissors into 1–2 mm pieces and homogenized by gentle pressing through a 150-μm mesh stainless-steel sieve with the bottom of a 250 ml conical flask in a rolling action. To prepare the sucrose gradients, place 6 ml of buffer D (0.86 M sucrose, 5 mM $MgCl_2$, 0.1 M phosphate buffer pH 6.7) in Beckman SW-41 Ultraclear tubes and overlay 5 ml of homogenate and 1 ml of buffer B (0.25 M sucrose, 5 mM $MgCl_2$, 0.1 M phosphate buffer pH 6.7). After centrifugation at $103,800 \times g$ (29,000 rpm) in an SW-41 rotor for 60 min at 4 °C, the lipid at the top was aspirated and the Golgi fractions accumulated at the 0.5/0.86 M sucrose interfaces were collected. The Golgi fractions were adjusted to 0.25 M sucrose concentration (refractive index, 1.3456) using buffer A (5 mM $MgCl_2$, 0.1 M phosphate buffer pH 6.7). Pooled

Golgi fractions were loaded onto the second gradient in the same centrifuge tube by adding 1 ml buffer E (1.3 M sucrose, 5 mM MgCl$_2$, 0.1 M phosphate buffer pH 6.7), 2 ml buffer C, and 9 ml diluted Golgi fractions. After centrifugation at 7900 × $g$ (8000 rpm) in an SW-41 rotor for 30 min, Golgi membranes concentrated at the 0.5 M/1.3 M sucrose interface were collected and gently mixed with 1 volume buffer A. Purified Golgi membranes were aliquoted and snap-frozen in liquid nitrogen and stored at −80 °C.

## Mammalian cell culture

Cell lines used in this study are listed in Supplementary Table 2. HEK293 (CRL-1573), HEK293T (CRL-3216), and HeLa (CCL-2) were purchased from ATCC. Cells were cultured in DMEM (Invitrogen) containing 10% Super Calf Serum (Gemini), 1% penicillin and strepto-mycin (Invitrogen), and 1 μg/ml plasmocin (Invivogen) at 37 °C, 5% CO$_2$. All cells were tested negative for mycoplasma.

## Plasmids

Plasmids used in this study are listed in Supplementary Table 3. The CDS of TMEM251 was purchased from DNASU plasmid Repository (Arizona State University). The CDS of GNPTAB is a generous gift from Dr. Stuart Kornfeld at the Washington University in St. Louis. The LIPA plasmid is a generous gift from Dr. Morihisa Fujita at Jiangnan University, China.

## Transfection

HEK293 cells were cultured in DMEM containing 10% serum-only media for at least 4 days before transfection. Cells were transfected with individual overexpression plasmids (2.4 μg DNA for a 3.5 cm dish) using Lipofectamine 2000 (Invitrogen) according to the manu-facturer's instructions.

## Generation of lentiviral stable cell lines

Stable cell lines were generated as described in Zhang et al.[17]. In Brief, HEK293T cells were transfected with transfer plasmid, psPAX2 (Addgene 12260), and pMD2.G (Addgene 12259) at a 3.5:3.5:1 ratio using Lipofectamine 2000 according to the manufacturer's instruc-tion. 72 h after transfection, the supernatant was collected and applied through a 0.45 μm filter. To generate stable cell lines, HEK293, HEK293T, or HeLa cells were seeded in 3.5 or 6 cm dishes and infected with the infectious media (DMEM containing 10% super calf serum, 10 μg/ml polybrene, MOI between 0.3 to 0.5). The puromycin selection was used at 1 μg/ml, and the blasticidin selection was used at 10 μg/ml. The selection lasted for at least 10 days before subsequent analysis.

## Generation of CRISPR-Cas9 KO and KI cell lines

*TMEM251*, *GNPTAB*, and *CI-MPR* knockout HEK293 or HeLa cells were generated as described in Ran et al.[77]. In brief, sgRNA guides were ligated into pspCas9(BB)−2A-Puro (Addgene, 48139) or Lenti-multi-CRISPR (Addgene 85402) plasmids. For single colonies, cells were transfected with CRISPR-Cas9 knockout plasmids using Lipofectamine 2000 according to the manufacturer's instruction. After 24 h of transfection, cells were treated with 1 μg/ml puromycin (Invitrogen) for 48 h. Single cells were isolated into 96-well plates using limited dilution to a final concentration of 0.5 cell per well. The knockout colonies were screened by western blot analysis. The KO cell lines were verified by sequencing analysis to confirm the indels at target sites. For polyclonal KO cell lines, cells were transduced with Lentivirus-based CRISPR-Cas9 plasmids. After 24 h, cells were treated with 1 μg/ml puromycin for 7 days.

To generate the template for GNPTAB knock-in (KI), 300 bp homology arms (upstream and downstream from the stop codon) were amplified from the genomic DNA. The 3HA coding sequence was inserted in between the homology arms by overlapping extension. The resulted DNA fragment was ligated into the pGEM-T Easy vector. To generate GNPTAB-3HA KI cells, HEK293T cells from a 6 cm dish were transfected with 4 μg of template plasmid and 2 μg CRISPR-Cas9 plas-mid using Lipofectamine 2000 according to the manufacturer's instruction. After 24 h of transfection, cells were treated with 1 μg/ml puromycin (Invitrogen) for 48 h. Single cells were isolated into 96-well plates using limited dilution to a final concentration of 0.5 cell per well. The knockin colonies were screened by PCR using a 3HA internal for-ward primer and a reverse primer located 600 bp downstream of the stop codon. The KI colonies were further verified by western blot and sequencing analysis.

The following reported sgRNAs were used in this study[22]:
TMEM251 sgRNA1: 5′-ATGAACTTCCGTCAGCGGAT-3′,
TMEM251 sgRNA2: 5′-TGTCCACACCCAAAAAGGCA-3′,
TMEM251 sgRNA3: 5′-ATAGTAAAATGCTGCTGCAC-3′,
GNPTAB sgRNA1: 5′-ACTCATTGCGATCTATCGAG-3′,
GNPTAB sgRNA2 (KI): 5′-CTTCTATACTCTGATTCGAT-3′,
CI-MPR sgRNA: 5′-GCTCAAAGATCCATTCGCCG-3′

## CRISPR-Cas9 knockout screen

The FACS-based CRISPR-Cas9 knockout screen was performed according to Joung et al. and Lenk et al.[78,79]. The human Brunello CRISPR knockout pooled library was purchased from Addgene (73178) and amplified according to manufactory instructions[22]. Lentiviral par-ticles were produced by the Vector Core at the University of Michigan.

HEK293 cells that stably express GFP-RNF152-IRES-mCherry and Cas9 were cultured in twenty 15 cm dishes to reach 50% confluency. Viruses containing DMEM were added to reach MOI = 0.25. After 24 h, cells were treated with 1 μg/ml puromycin for 7 days.

About $1.5 \times 10^8$ Transduced cells were subjected to FACS using FACSAria III cell sorter (BD Biosciences). The top 1–1.5% cells with a high GFP/mCherry ratio were collected. About $6 \times 10^5$ of such events/cells were captured, plated, and expanded for 18 days. About $10^8$ cells were subjected to a second round of FACS, and $3.2 \times 10^5$ events/cells were captured, plated, and expanded for 7 days. Genomic DNA was purified from the expanded population after each round of sorting as well as from the initial transduced population, using the Gentra Purogene kit (Qiagen). Integrated sgRNAs were enriched by PCR amplification, with eight replicate PCR reactions each with 1 μg template per reaction, in order maintain a complex sampling of the cellular population. Adapter sequences and per-sample barcodes were added to libraries by a sec-ond round of PCR. The libraries were pooled and sequenced on an Illumina MiSeq instrument using 150-bp single-end reads. Constant adapter sequences were trimmed from the resulting reads with Cutadapt[80], and gene-level enrichment scores and false discovery rate (FDR) estimates were calculated using the CB2 package in R[81].

## Sample preparation and western blotting

Cells were collected in ice-cold 1X PBS, pelleted at 2700 × $g$ for 2 min, and lysed in lysis buffer (20 mM Tris pH = 8.0, 150 mM NaCl, 1% Triton) containing protease inhibitor cocktail (Bimake) at 4 °C for 20 min. Cell lysates were centrifuged at 18,000 × $g$ for 15 min at 4 °C. The protein concentration of the supernatant was measured by Bradford assay (Bio-rad) and normalized. After adding 2X urea sample buffer (150 mM Tris pH 6.8, 6 M Urea, 6% SDS, 40% glycerol, 100 mM DTT, 0.1% Bro-mophenol blue), samples were heated at 65 °C for 10 min. 30 μg of each lysate was loaded and separated on SDS-PAGE gels. Note that for the TMEM251 (GCAF) blot in Figs. 6b, 7b, c, and Supplementary Fig. 5d, only 1/20 of the samples were loaded in the GCAF overexpression lanes. Protein samples were transferred to a nitrocellulose membrane for western blot analysis. After incubated with primary and secondary antibodies, membranes were scanned using the Odyssey CLx imaging system (LI-COR) or developed with CL-XPosure film (Thermo Scientific).

The following primary antibodies were used for western blotting in this study: rabbit anti-GFP (1:3000, TP401, Torrey Pines Biolabs), mouse anti-actin (1:5000, 66009-1-lg, Proteintech), mouse anti-GAPDH

(1:2000, 60004-1-1g, Proteintech), rabbit anti-CTSD (1:1000, 21327-1-AP, Proteintech), rabbit anti-Golgin160 (1:1000, 21193-1-AP, Proteintech), rabbit anti-p62 (1:2000, 18420-1-AP, Proteintech), rabbit anti-LC3 (1:2000, 14600-1-AP, Proteintech), rabbit anti-IGF2R (CI-MPR) (1:2000, 20253-1-AP, Proteintech), mouse anti-HA (1:500, 16B12, BioLegend), mouse anti-CTSC (1:500, D-6, Santa Cruz Biotechnology), mouse anti- SREBF2/SREBP2 (1:500, 1C6, Santa Cruz Biotechnology), rabbit anti-FLAG (1:2000, H6908, Millipore-Sigma), rabbit anti-LAPTM4A (1:1000, HPA068554-1, Millipore-Sigma), rabbit anti-TMEM251 (1:1000, HPA-48559, Millipore-Sigma), mouse anti-V5 (1:3000, 46-0705, Invitrogen), rabbit anti-ATF6 (1:1000, 24169-1-AP, Proteintech), rabbit anti-EGFR (1:2000, a generous gift from Dr. Stuart Decker at the University of Michigan).

The plasmid for single-chain antibody against M6P (scFv M6P) was purchased from the Geneva Antibody Facility (AG949, University of Geneva). The full construct contains an N-terminal IL-2 signal sequence and a C-terminal Fc region from the rabbit IgG. To produce scFv against M6P, HEK293T cells were transfected with AG949 plasmid. After 48 h, cells were washed with serum-free DMEM and incubated with serum-free DMEM for 24 h. The supernatant is filtered with a 0.45 μm filter. This filtered supernatant is directly used as a primary antibody (without dilution) to detect M6P.

The following secondary antibodies were used in this study: goat anti-mouse IRDye 680LT (926-68020), goat anti-mouse IRDye 800CW (926-32210), goat anti-rabbit IRDye 680LT (926-68021), goat anti-rabbit IRDye 800CW (926-32211). These secondary antibodies were purchased from LI-COR Biosciences and used at 1:10,000 dilution.

To detect TMEM251, M6P (scFv M6P), or GNPTAB-3xHA KI, the anti-protein A HRP (PA00-03, Rockland, for TMEM251 and M6P) or mouse HRP (115-035-046, Jackson labs, for GNPTAB-3xHA KI) secondary antibodies were used at 1:10,000 dilution. The signal was detected with the Pierce ECL kit (Thermo Scientific).

## EGFR degradation assay

HeLa cells were cultured to 70–80% confluency in 6 cm dishes. Cells were washed with serum-free DMEM twice and incubated with serum-free DMEM. After 14 h, 100 ng/ml of EGF (Invitrogen) was added to cells. Cells were collected in ice-cold PBS at the indicated time, pelleted at $2700 \times g$ for 2 min, and stored at −80 °C before subsequent western blot analysis.

## Membrane isolation

The membrane isolation protocol was adapted from Shao and Espenshade[82], with some modifications. Cells with 70–80% confluency from a 10 cm dish were collected in ice-cold 1X PBS, pelleted at $2700 \times g$ for 2 min. The pelleted cells were resuspended in 1 ml ice-cold membrane isolation buffer (1 mM EDTA and 1 mM EGTA in 1X PBS, with protease inhibitor) and homogenized. The homogenate was centrifuged at $900 \times g$ for 5 min at 4 °C, and the supernatant was transferred to a new tube and centrifuged at $20,000 \times g$ for 20 min at 4 °C to collect membranes. After centrifugation, the membrane pellet was further dissolved in lysis buffer (20 mM Tris pH = 8.0, 150 mM NaCl, and 1% Triton) containing 1X protease inhibitor cocktail (Biomake) at 4 °C for 20 min. The undissolved membranes were removed by another round of centrifugation at $20,000 \times g$ for 15 min at 4 °C, and the protein concentration from the supernatant was measured by Bradford assay and normalized. Samples were incubated with 2X urea sample buffer samples at 65 °C for 8 min before western blot analysis.

## SREBF2/SREBP2 processing assay

The SREBF2/SREBP2 processing assay was adapted from Shao and Espenshade[82], with some modifications. Cells were cultured to 50% confluency, treated with 50 μM sodium compactin (Millipore-Sigma) and 50 μM sodium mevalonate (Millipore-Sigma) in the presence or absence of sterols (1 μg/ml 25-Hydroxycholesterol [25-HC], Millipore-Sigma), 10 μg/ml cholesterol (Millipore-Sigma). After 16 h, N-acetyl-leucinyl-leucinyl-norleucinal (ALLN, Millipore-Sigma) was added to a final concentration of 25 μg/ml, and cells were harvested 1 h later for membrane isolation and western blot analysis.

## CI-MPR binding assay

Cells with 70–80% confluency in a 10 cm dish were washed twice with serum-free DMEM and then incubated with serum-free DMEM for secretion. After 16 h, conditioned media was collected and transferred to a 50 ml conical tube. The media was centrifuged at $500 \times g$ for 5 min to remove cell debris, filtered with a 0.45 μm filter, and concentrated to ~200 μl using 10 kDa cutoff Amicon Centrifugal filters (Millipore-Sigma). The protein concentration from the concentrated media was measured by Bradford assay and normalized. After adding 2X urea sample buffer, samples were heated at 65 °C for 8 min and loaded onto SDS-PAGE gel for western blot analysis. After transfer, the nitrocellulose membrane was blocked with 3% BSA, and incubated with biotinylated CIMPR protein (0.25 μg/ml in 3% BSA, a generous gift from Dr. Peter Lobel, Rutgers University) as the primary binder at 4 °C. After 14 h incubation, the membrane was further incubated with Streptavidin secondary antibodies (IRDye® 800CW Streptavidin, 926322230, LI-COR Biosciences) and scanned using the Odyssey CLx imaging system.

## Rescue of TMEM251 KO cells with conditioned media

CI-MPR KO and GNPTAB KO cells were cultured to reach 70–80% confluency. Cells were washed twice with serum-free DMEM and then incubated with serum-free DMEM for secretion. After 16 h, the conditioned media from different cell lines were collected and transferred to a 50 ml conical tube. The media was centrifuged at $500 \times g$ for 5 min to remove cell debris, filtered with a 0.45 μm filter, and concentrated to ~500 μl using 10 kDa cutoff Amicon Centrifugal filters (Millipore-Sigma). The concentrated media were added to TMEM251KO cells (~5% confluency, 4 ml complete media in 6 cm dishes). For the mannose-6-phosphate (M6P) competition experiment, 10 mM of M6P (Millipore-Sigma) was added to the TMEM251KO cells 3 h before the addition of concentrated conditioned media from the CI-MPR KO cells.

During cell growth, new conditioned media from fresh CI-MPR KO and GNPTAB KO cells were concentrated every the other day and fed to TMEM251 KO cells. For the M6P competition dish, TMEM251 cells were always pre-treated with 10 mM M6P for 3 h before adding the new conditioned media. After 7 days, cells were harvested for analysis.

## Secretome analysis

HEK293 WT and sgTMEM251 cells were cultured in 15 cm dishes to reach 70–80% confluency. Cells were washed with serum-free DMEM three times and incubated with 20 ml serum-free DMEM for 14 h. The conditioned media were collected and transferred to a 50 ml conical tube. The media was centrifuged at $500 \times g$ for 5 min to remove cell debris, filtered with a 0.45 μm filter, and concentrated to ~200 μl using 3 kDa cutoff Amicon Centrifugal filters (Millipore-Sigma). The protein concentration from the conditioned media was measured by Bradford assay and normalized. Samples were mixed with 2XUrea buffer and heated at 65 °C for 8 min.

About ~70 μg of protein samples were loaded onto an SDS-PAGE gel and run for 4.5 cm into the gel. Samples were stained with Sypro Ruby gel stain (Invitrogen) and excised for MS analysis. The Mass spectrometry (MS) analysis was performed by the Taplin Mass Spectrometry Facility at the Harvard Medical School.

Statistically significant proteins were determined as having absolute log-fold change larger than 2 and a $p$-value < 0.05. Gene ontology enrichment analyses were performed using Metascape, a web-based biological annotation database[83].

## Immunostaining, microscopy, and image processing

Immunostaining was performed as described[17], with some modifications. Cells grown on 1.5 circular glass coverslips were washed with ice-cold 1X PBS and fixed in 4% paraformaldehyde for 10 min at room temperature. Cells were permeabilized with 0.3% Triton in PBS for 15 min. For immunostaining of LAMP2, cells were fixed and permeabilized in cold 100% methanol for 8 min at −20 °C. The samples were blocked in 3% BSA (in 1XPBS) for 30 min at room temperature, followed by incubating with primary and secondary antibodies. The cell nucleus was stained using Hoechst (1:8000, Invitrogen). Coverslips were mounted in Fluoromount-G (SouthernBiotech) and cured for 24 h before imaging.

The following primary antibodies were used for immunostaining in this study: mouse anti-LAMP2 (1:100, H4B4, DHSB), rabbit anti-TMEM251 (1:100, HPA-48559, Millipore-Sigma), anti-EEA1 (1:50, sc-137130, Santa Cruz Biotechnology), mouse anti-GM130 (1:200, 610822, BD Biosciences), mouse anti-HA (1:50, 16B12, BioLegend), rabbit anti-GPP130 (1:100, PRB-144C, BioLegend).

The following secondary antibodies were used at 1:100: FITC goat anti-rabbit (111-095-003, Jackson ImmunoReseach) and TRITC goat anti-mouse (115-025-003, Jackson ImmunoReseach).

Microscopy was performed with a DeltaVision system (GE Healthcare Life Sciences). The DeltaVision microscope was equipped with a scientific CMOS camera and an Olympus UPLXAP0100X objective. The following filter sets were used: FITC (excitation, 475/28; emission, 525/48), TRITC (excitation 542/27; emission 594/45), and DAPI (excitation 390/18; emission 435/48). Image acquisition and deconvolution were performed with the SOFTWORX program. Images were further cropped or adjusted using ImageJ (National Institutes of Health).

## Immunoprecipitation

Immunoprecipitation was performed 48 h post-transfection according to the manufacturer's instruction with some modifications. In brief, cells (one 15 cm dish of near-confluent cells per IP group) were collected in ice-cold 1XPBS, pelleted at $2700 \times g$ for 2 min, and lysed in 1 ml of lysis buffer (20 mM Tris pH = 8.0, 150 mM NaCl, 1% Triton) containing protease inhibitor cocktail (Biomake) at 4 °C for 20 min. Cell lysates were centrifuged at $18,000 \times g$ for 15 min at 4 °C. The concentration of the supernatant was measured by Bradford assay (Bio-rad) and normalized. 30 µl beads (pre-equilibrated with lysis buffer, anti-FLAG M2 beads, Millipore-Sigma; anti-V5 beads, Invitrogen; anti-HA beads, ThermoFisher) were added to the normalized cell lysate and incubated at 4 °C overnight (2 hrs for FLAG M2 beads) with gentle rocking. The resin was then washed 4 times with lysis buffer. For FLAG IP, the bound proteins on the anti-FLAG M2 beads were eluted with 3xFLAG peptides and precipitated by 10% TCA precipitation for 1 h. The pellet was washed with 0.1% TCA, resuspended with 2X Urea sample buffer (150 mM Tris pH 6.8, 6 M Urea, 6% SDS, 40% glycerol, 100 mM DTT, 0.1% Bromophenol blue). The sample was treated with bead beating for 10 min and heated at 65 °C for 10 min. For V5 and HA IP, after washing, 2X Urea sample buffer was directly added to the resin and heated at 65 °C for 10 min. The resulting eluates were analyzed by western blot.

## Lyso-IP

Lyso-IP was conducted as described before[17,84]. Briefly, ~8 × 10⁶ HEK293T cells that stably expressed TMEM192-3HA or TMEM192-2FLAG were collected in ice-cold PBS. ~2.5% of the cells were used as input and further processed for western blot. The rest of the cells were spun down at $1000 \times g$ for 2 min, resuspended with ice-cold PBS containing protease inhibitor cocktail, and homogenized. The samples were then centrifuged at $1000 \times g$ for 2 min. The supernatant was incubated with 20 µl of anti-HA magnetic beads for 20 min at 4 °C. The beads were washed with PBS five times and then heated at 65 °C for 10 min in a 2xUrea sample buffer (150 mM Tris pH 6.8, 6 M Urea, 6%

SDS, 40% glycerol, 100 mM DTT, 0.1% Bromophenol blue). Samples were further analyzed by western blot.

## RNA-sequencing

Total RNA samples were extracted from either WT or 251 KO HEK293 cells using TRIzol (Thermo Fisher Scientific) and the PureLink RNA Mini Kit (Invitrogen). For each sample (three WT and three TMEM251 KO using different sgRNAs), around 3 µg of total RNA was submitted to the Advanced Genomics Core at the University of Michigan. After quality control, the mRNAs from total RNAs were enriched with a poly-A based selection method prior to cDNA synthesis, and the sequencing was then performed on the NovaSeq with 150 bp paired end reads (PE150) to target 30–40 million reads/sample.

The raw reads were filtered using RSeQC with default parameters by removing low-quality bases (>Q30) and adapter-contaminated reads. The resulting high-quality clean reads in fastq format were trimmed using Trim Galore (v 0.5.0) and aligned to the human genome (Sequence: ENSEMBL-GRCh38) using STAR (v 2.6.0)[85] After mapping, read counts were generated by HTSeq-count (v.0.11.3)[86]. The read counts were used for a differential expression analysis between wild-type control cells and TMEM251 knockout cells using R package DESeq2 (v.1.28.1)[87]. Statistically significantly expressed genes were determined as having absolute log-fold change larger than 1.2 and a p-value < 0.05 based on the Benjamini-Hochberg procedure, which controls the false discovery rate (FDR). Principal component analysis (PCA) and heatmaps of differentially expressed genes (DEGs) were generated using ClustVis (https://biit.cs.ut.ee/clustvis/). DEGs were processed for gene ontology enrichment analyses using Metascape.

## Lysotracker staining and flow cytometry analysis

Cells were treated with 50 nM lysotracker Red DND-99 (Thermo Fisher Scientific) for 30 min. Cells were washed with 1XPBS and trypsinized until all cells were dissociated from the dishes. Dissociated cells were neutralized with DMEM containing 10% serum media and pelleted at $300 \times g$ for 3 min. Cells were resuspended in ice-cold 1XPBS and analyzed by a Ze5 (Bio-rad) flow cytometer. The data were analyzed using FlowJo software.

## Transmission electron microscopy

HeLa cells were cultured on an 8 mm (diameter) Thermanox coverslip to 80–90% confluency and processed as cell monolayer without any modifications. The cells were pre-fixed in 1.25% glutaraldehyde in 0.05 M cacodylate buffer at 37 °C for 5 min and further at 4 °C overnight, followed by post-fixed in a mixture of 1% osmium tetroxide (OsO4) plus 1% potassium ferrocyanide [K4Fe(CN)6] in 0.1 M cacodylate buffer. To better contrast the cell and subcellular membranes, the pre- and post-fixed cells were stained with 1% thiocarbohydrazide, 1% OsO4, 1% uranyl acetate, and Walton's lead aspartate. The cells were dehydrated in ascending ethanol series (10, 30, 50, 70, 80, 90, 95, 100%) and infiltrated in Durcupan resin. The resin infiltrated cells were thermally polymerized at 70 °C for 48 h. 70 nm ultrathin sections were cut by a Leica EM UC7 ultramicrotome and the sections were placed on a 300 mesh Cu bare grid. The ultrathin sections were coated with 4 nm carbon by a Leica EM ACE600 high vacuum coater. The ultrathin sections were observed under a JEOL JEM-1400 Plus LaB6 transmission electron microscope at 60 keV high tension and imaged by an AMT NanoSprint 12 megapixel CMOS camera.

## Alcian blue and Alizarin red staining

Alizarin red staining was performed following a published protocol[88]. Briefly, 7 dpf fish embryos were fixed by 4% PFA overnight at 4 °C. Embryos were then dehydrated in 50% ethanol for 10 min and bleached by 1% H₂O₂/0.5% KOH for 10 min. Embryos were stained with 0.04 mg/ml Alizarin Red in 1% KOH for 30 min and stored in 50% glycerol/0.25% KOH until imaging.

Alcian blue staining was previously described[50]. Briefly, 4 dpf fish embryos were fixed by 4% PFA overnight at 4 °C. Embryos were then dehydrated in 50% ethanol for 10 min and stained with 0.02% Alcian blue, 200 mM MgCl$_2$ in 70% ethanol for 3 h. Embryos were bleached in 1.5% H$_2$O$_2$/1% KOH for 30 min, cleaned in 20% glycerol/0.25% KOH overnight, and stored in 50% glycerol/0.25% KOH until imaging.

Images were captured with a stereomicroscope (Leica MZ16F) equipped with a QImaging QICAM camera.

### The F0 knockout of TMEM251 and GNPTAB in zebrafish
Cas9 mRNA was synthesized by in vitro transcription using the pT3.Cas9-UTRglobin plasmid (a kind gift from Prof. Yonghua Sun from the Institute of Hydrobiology, Chinese Academy of Sciences) as the template. Four sgRNAs targeting TMEM251 were designed using CHOPCHOP[89]. The primers used to synthesize gRNAs that target TMEM251 and GNPTAB are listed in Supplementary Table 4. The sgRNAs were synthesized by in vitro transcription following a published method[51]. Once synthesized, the sgRNAs (40 ng/µl) were mixed with Cas9 mRNA (400 ng/µl) and co-injected into WT embryos at the one-cell stage. The injected embryos were raised in E3 embryo medium, with PTU added at 1 dpf.

### Quantification and statistical analysis
The band intensity for western blot was quantified using Image Studio software (LI-COR). The rate constants (k) of GFP-RNF152, LAPTM4A and EGFR were calculated by fitting the data to the first-order decay and the rate constant in Excel. The half-lives were calculated by $t_{(1/2)} = \ln2/k$. Graphs were generated using Prism (GraphPad). Statistical analysis was performed with the two-tailed unpaired t-test (Figs. 2j–k; 3e; 4b, c) or one-way ANOVA (Figs. 2b, c, e, f, h; 3f, g; 5i–l; 6c, f, i; 7e, f; Supplementary Figs. 1d, e; 3c, d; 5e; Supplementary Table 1; Supplementary Data 1–3). Error bars represent the standard deviation. *≤0.05, **≤0.01, ***≤0.001, ****≤0.0001.

### Reporting summary
Further information on research design is available in the Nature Research Reporting Summary linked to this article.

## Data availability
Data supporting this study are provided within the paper and supplementary files. The RNA-seq data generated from this study has been uploaded to NCBI with accession code GSE209652. The processed raw data is attached as Supplementary Data 2. The secretome data generated from this data is attached as Supplementary Data 3. Source data are provided with this paper.

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

## Acknowledgements

We thank the Li laboratory members for their helpful discussion and technical support. We are grateful to Drs. S. Kornfeld, B. Doray, P. Lobel, M. Fujita for sharing reagents. This research is supported by the MCubed 3.0 fund, the Protein Folding and Diseases Initiative, and a MICHR Pathway Pilot grant from the University of Michigan, and NIH grants R01GM133873 and GM133873-01S2 to M.L., R01GM129123 to J.K., R35GM130331 to Y.W. Research in C. Duan's lab is supported by NSF grants IOS-1557850 and IOS-1755268.

## Author contributions

Conceptualization: W.Z., X.Y., and M.L.; methodology: W.Z., X.Y., and M.L.; investigation: W.Z., X.Y., Y.L., L.Y., B.Z., W.C., V.V., L.C., and B.B.; some critical reagents: J.Z., S.B., and Y.W.; writing & editing: W.Z., X.Y., and M.L.; funding acquisition: M.L., Y.W., C.D., and J.K.; resources & supervision: M.L. W.Z., and X.Y. contributed equally, and both have the right to put themselves first in bibliographic documents.

## Competing interests

The authors declare no competing interests.

## Additional information

**Supplementary information** The online version contains

supplementary material available at https://doi.org/10.1038/s41467-022-33025-1.

