## [Peer Review File · Nature Communications]

GCAF(TM251) regulates lysosome biogenesis by activating the mannose-6-phosphate pathwayEditorial Note: Parts of this Peer Review File have been redacted as indicated to maintain the confidentiality of unpublished data.

REVIEWER COMMENTS

Reviewer #1 (Remarks to the Author):

The manuscript by Zhang and colleagues explores the rapid lysosomal degradation as the basis for a screen for lysosomal regulators. They identified several known lysosomal regulators, such as vATPase subunits, and TMEM251. This protein had already been implicated in lysosomal function, but the mechanism was unclear.

Here, the authors showed that TMEM251 was involved in the mannose-6-P pathway that targets proteins to lysosomes. When TMEM251 is absent, the luminal proteins do not reach the lysosomes and therefore the lysosomes are dysfunctional. The authors further show that this phenotype can be rescued by providing M6P-associated proteins from WT cell extracts that can be internalized by TMEM251-KO cells and locate to the lysosomes.

The study is very well done from a technical point of view, and the authors pre-emptively addressed possible weaknesses. The only aspect that seems somewhat neglected is the possible effect of TMEM251 in lysosomal membrane proteins.

There are also a few aspects that would benefit from further clarification:

- why the need for two rounds of screening?
- why using HeLa for the EGF experiment when HEK was used for everything else?
- in page 9, it is important to clarify the difference between CI-MPR and CD-MPR for the non-expert audience
- similarly, S1P needs a better context for non-experts
- lane 241 "compare" is misspelled

I cannot comment on the zebrafish technical aspects because that is outside of my expertise area. The phenotypes observed in the animal studies were in accordance with the cell line results.

Overall, the experiments are well carried out, the data is convincing, the necessary controls are in place, and the conclusions are well aligned with the experiments.

Reviewer #2 (Remarks to the Author):

The manuscript by Zhang et al reports the identification of GCAF, a Golgi-localized transmembrane protein formerly annotated as TMEM251, as a new essential factor for lysosomal hydrolase delivery. Building on their previous finding that certain lysosomal proteins are subjected to rapid turnover, the authors set up a genome-wide CRISPR-Cas9 screens aimed at identifying factors required for lysosomal degradation of a dual-fluorescent reporter. Along with expected hits, the screen yields TMEM251/GCAF as an uncharacterized gene, in the absence of which lysosomal proteolysis is severely impaired. Through follow-up mechanistic investigation, the authors find that GCAF is not required for lysosomal acidification but, rather, for delivery of numerous hydrolases to the lysosome.

Mechanistically, they show that GCAF bridges GlcNAc-1-phosphate transferase (GNPT) to the site-1 protease (S1P), thus allowing proteolytic activation of the former by the latter. The authors go on to show that GCAF knock down in zebrafish results in severe developmental delay and dysplasia, phenotypic traits that resemble the lysosomal disorder Mucopolysaccharidosis type II. Given the genetic, mechanistic and phenotypic similarities between Mucopolysaccharidosis type II and loss of TMEM251, and a recently reported human disease caused by germline mutations in the TMEM251 gene, it is concluded that these mutations underlie a novel human lysosomal disease, tentatively named Mucopolysaccharidosis type V.

Overall, this is a superbly executed study, clearly written and supported by compelling, high-quality data, which is set to significantly advance our understanding of lysosomal biogenesis and function. The screen is cleverly designed and the hits make perfect biological sense. The follow-up mechanistic investigation is based on rigorous microscopy and biochemistry and leaves little doubt as to the mechanistic aspects of TMEM251 function. Finally, the zebrafish experiments add pathophysiological relevance to the findings.

I do not see any need for changes to the manuscript content or structure, and I recommend speedy

acceptance and publication in Nature Communications.

Reviewer #3 (Remarks to the Author):

This is an interesting study that provides convincing evidence that the membrane protein TMEM-251 is essential for the functioning of GlcNAc-phosphotransferase (GNPT) in the Golgi. GNPT mediates the phosphorylation of newly synthesized lysosomal enzymes a step that is required for their transport to lysosomes.

The discovery emerged from a clever genome-wide CRISPR/Cas9 knockout screen to identify genes essential for lysosomal function. The key was to generate mutants that failed to degrade a lysosomal membrane protein that was constitutively degraded. The screen identified 27 genes that were required for degradation of the membrane protein. Of these, the authors focused on TMEM-251 as its function was unknown at the time.

The authors carried out an extensive analysis of cells lacking TMEM-251 revealing that there is hypersecretion of multiple lysosomal enzymes that fail to bind to the CI-MPR, indicating the lack of M6P residues. This explains the lysosomal dysfunction observed with TMEM-deficient cells. They show that TMEM-251 KO cells lack processed GNPT, which is the active form of the enzyme, explaining why the KO cells hypersecrete non-phosphorylated lysosomal enzymes. In co-IP experiments they demonstrate that TMEM-251 interacts with both GNPTAB precursor and with Site-1 protease, the enzyme that cleaves the GNPTAB precursor to form the active enzyme.

Finally they show that TMEM-251 deficiency in zebrafish results in a picture similar to GNPT deficiency in that organism.

Critique:

This is a well-done study that provides convincing evidence that the membrane protein TMEM-251 is essential for the expression of the active form of GNPT in the Golgi. In the absence of TMEM-251, the ability to phosphorylate lysosomal enzymes is lost, hypersecretion of these enzymes occurs and lysosomal dysfunction results. In 2021 several patients from two families with TMEM-251 mutations who presented with a MLII phenotype as occurs with GNPTAB deficiency were reported, consistent with the current findings (Ref.10).

The weakness of the study is that the mechanism whereby the TMEM-251 facilitates the activation of GNPTAB is not established. In this regard, the authors could perform IF analysis on the GNPTAB-3xHA(KI)/TMEM-251 KO cells (Fig.6B) to determine the intracellular localization of the GNPTAB. Is the GNPTAB in the Golgi or does it require TMEM-251 to exit the ER? This would serve to localize the site of the interaction. (This cell line could also be used to establish whether the TMEM-251 missense mutations reported in the patients (Ref10) are inactive as predicted or not). The authors do show that TMEM-251 binds to both GNPTAB and S1P, but the basis for these interactions is not established. Overall, the reported findings are quite novel and highly significant on their own and warrant publication in Nature Communications.

Reviewer #1 (Remarks to the Author):

The manuscript by Zhang and colleagues explores the rapid lysosomal degradation as the basis for a screen for lysosomal regulators. They identified several known lysosomal regulators, such as vATPase subunits, and TMEM251. This protein had already been implicated in lysosomal function, but the mechanism was unclear.

Here, the authors showed that TMEM251 was involved in the mannose-6-P pathway that targets proteins to lysosomes. When TMEM251 is absent, the luminal proteins do not reach the lysosomes and therefore the lysosomes are dysfunctional. The authors further show that this phenotype can be rescued by providing M6P-associated proteins from WT cell extracts that can be internalized by TMEM251-KO cells and locate to the lysosomes.

The study is very well done from a technical point of view, and the authors pre-emptively addressed possible weaknesses. The only aspect that seems somewhat neglected is the possible effect of TMEM251 in lysosomal membrane proteins.

We thank this reviewer for the very positive comments. We used LAPTM4A and RNF152 to confirm that the degradation of lysosome membrane proteins (LMPs) is impaired after knocking out GCAF. The degradation of human LMPs is a very novel topic. So far, the only known fast-degrading LMPs are RNF152 and LAPTM4A. Both of them have been tested by this study. How TMEM251 might affect the trafficking of LMPs is an exciting question but beyond the scope of the current study. It will be pursued by our future studies.

There are also a few aspects that would benefit from further clarification:
- why the need for two rounds of screening?

Based on our experience¹, one additional round of sorting can further enrich the positive population. As shown in this manuscript, the results from two rounds of screening are largely consistent, but hits from the second round of screening have a further increase in log₂FC and FDR. For example, the log₂FC values of TMEM251 were 5.43 for the first round and 9.52 for the second round. Please refer to supplemental table 1 for more details.

- why using HeLa for the EGF experiment when HEK was used for everything else?

We used HEK293 cells for most experiments because the original CRISPR screen was done using HEK293 cells. However, endogenous EGFR expression level is very low in HEK293 cells^{2,3}. This is why we used HeLa cells for the EGFR endocytosis experiment, which has a high expression level. We revised the sentence in the main text for clarification.

- in page 9, it is important to clarify the difference between CI-MPR and CD-MPR for the non-expert audience

- similarly, S1P needs a better context for non-experts

We thank this reviewer for the excellent suggestion. In the revised manuscript, we have introduced more background information about S1P, CI-MPR, and CD-MPR. Please see page 9 for details.

- lane 241 "compare" is misspelled

We corrected it, thank you!

I cannot comment on the zebrafish technical aspects because that is outside of my expertise area. The phenotypes observed in the animal studies were in accordance with the cell line results.

The Zebrafish experiments were carried out in collaboration with Dr. Cunming Duan, a world expert on Zebrafish development.

Overall, the experiments are well carried out, the data is convincing, the necessary controls are in place, and the conclusions are well aligned with the experiments.

We thank the reviewer for the time and effort in reviewing our manuscript, and we appreciate the very positive feedback and suggestions.

Reviewer #2 (Remarks to the Author):

The manuscript by Zhang et al reports the identification of GCAF, a Golgi-localized transmembrane protein formerly annotated as TMEM251, as a new essential factor for lysosomal hydrolase delivery. Building on their previous finding that certain lysosomal proteins are subjected to rapid turnover, the authors set up a genome-wide CRISPR-Cas9 screens aimed at identifying factors required for lysosomal degradation of a dual-fluorescent reporter. Along with expected hits, the screen yields TMEM251/GCAF as an uncharacterized gene, in the absence of which lysosomal proteolysis is severely impaired. Through follow-up mechanistic investigation, the authors find that GCAF is not required for lysosomal acidification but, rather, for delivery of numerous hydrolases to the lysosome.

Mechanistically, they show that GCAF bridges GlcNAc-1-phosphate transferase (GNPT) to the site-1 protease (S1P), thus allowing proteolytic activation of the former by the latter. The authors go on to show that GCAF knock down in zebrafish results in severe developmental delay and dysplasia, phenotypic traits that resemble the lysosomal disorder Mucopolysaccharidosis type II. Given the genetic, mechanistic and phenotypic similarities between Mucopolysaccharidosis type II and loss of TMEM251, and a recently reported human disease caused by germline mutations in the TMEM251 gene, it is concluded that these mutations underlie a novel human lysosomal disease, tentatively named Mucopolysaccharidosis type V.

Overall, this is a superbly executed study, clearly written and supported by compelling, high-quality data, which is set to significantly advance our understanding of lysosomal biogenesis and function. The screen is cleverly designed and the hits make perfect biological sense. The follow-up mechanistic investigation is based on rigorous microscopy and biochemistry and leaves little doubt as to the mechanistic aspects of TMEM251 function. Finally, the zebrafish experiments add pathophysiological relevance to the findings.

I do not see any need for changes to the manuscript content or structure, and I recommend speedy acceptance and publication in Nature Communications.

We thank reviewer #2 for the time and effort in reviewing our manuscript, and we appreciate the very positive feedback.

Reviewer #3 (Remarks to the Author):

This is an interesting study that provides convincing evidence that the membrane protein TMEM-251 is essential for the functioning of GlcNAc-phosphotransferase (GNPT) in the Golgi. GNPT mediates the phosphorylation of newly synthesized lysosomal enzymes a step that is required for their transport to lysosomes.

The discovery emerged from a clever genome-wide CRISPR/Cas9 knockout screen to identify genes essential for lysosomal function. The key was to generate mutants that failed to degrade a lysosomal membrane protein that was constitutively degraded. The screen identified 27 genes that were required for degradation of the membrane protein. Of these, the authors focused on TMEM-251 as its function was unknown at the time. The authors carried out an extensive analysis of cells lacking TMEM-251 revealing that there is hypersecretion of multiple lysosomal enzymes that fail to bind to the CI-MPR, indicating the lack of M6P residues. This explains the lysosomal dysfunction observed with TMEM-deficient cells. They show that TMEM-251 KO cells lack processed GNPT, which is the active form of the enzyme, explaining why the KO cells hypersecrete non-phosphorylated lysosomal enzymes. In co-IP experiments they demonstrate that TMEM-251 interacts with both GNPTAB precursor and with Site-1 protease, the enzyme that cleaves the GNPTAB precursor to form the active enzyme. Finally they show that TMEM-251 deficiency in zebrafish results in a picture similar to GNPT deficiency in that organism.

Critique:

This is a well-done study that provides convincing evidence that the membrane protein TMEM-251 is essential for the expression of the active form of GNPT in the Golgi. In the absence of TMEM-251, the ability to phosphorylate lysosomal enzymes is lost, hypersecretion of these enzymes occurs and lysosomal dysfunction results. In 2021 several patients from two families with TMEM-251 mutations who presented with a MLI phenotype as occurs with GNPTAB deficiency were reported, consistent with the current findings (Ref. 10).

The weakness of the study is that the mechanism whereby the TMEM-251 facilitates the activation of GNPTAB is not established. In this regard, the authors could perform IF analysis on the GNPTAB-3xHA (KI)/TMEM-251 KO cells (Fig.6B) to determine the intracellular localization of the GNPTAB. Is the GNPTAB in the Golgi or does it require TMEM-251 to exit the ER? This would serve to localize the site of the interaction.

We completely agree with the reviewer for this constructive suggestion. To study the localization of endogenous GNPTAB, we have performed the immunofluorescence assay on GNPTAB-3xHA KI cells; we also built two independent GNPTAB-mNeonGreen KI lines using HeLa cells. However, despite many attempts, the endogenous GNPTAB expression level is too low to be detected by imaging, even though the GNPTAB-3HA KI line can be used for Western blotting purposes.

Therefore, following literature^{4,5}, we overexpressed GNPTAB-2HA in WT and TMEM251 KO cells to see if deleting TMEM251 will affect its ER exit. As shown in **supporting figure 1**, the processing of GNPTAB-2HA was also defective in TMEM251 KO cells, which is consistent with the results obtained from KI strains. Importantly, GNPTAB is localized to Golgi in both WT and TMEM251 KO cells, suggesting TMEM251 is not required for its ER exit. We have incorporated this result into the manuscript (Fig. S4F-G), and edited the text accordingly.

(This cell line could also be used to establish whether the TMEM-251 missense mutations reported in the patients (Ref10) are inactive as predicted or not).

We agree that TMEM251 patient variants need a closer investigation and will be one direction of our future studies. Focusing on R7W, we expressed the patient variant⁶ in TM251 KO cells to study if it is defective in M6P biogenesis and lysosome function. Surprisingly, under overexpression conditions, the R7W mutant largely complemented the lysosome defects, as evidenced by the rescue of mCTSD, LAMPTM4A, and LC3B-II levels (**supporting figure 2A**). This result suggested this missense mutant is still functional when overexpressed. And the phenotypes observed in patients are likely due to a protein level issue.

Consistent with this hypothesis, we noticed that under the same promoter, the steady-state protein level of the R7W mutant was always lower than the wild type TMEM251 (**supporting figure 2A**). It was also quickly degraded during a cycloheximide chase experiment (**supporting figure 2B**), confirming that R7W has a protein stability issue.

[redacted]

The authors do show that TMEM-251 binds to both GNPTAB and S1P, but the basis for these interactions is not established.

We completely agree that further mapping out the interaction domains will provide more mechanistic insights into how TMEM251 regulates the processing and activation of GNPTAB. But this will be beyond the scope of the current study. Adding to the difficulty of mapping binding sites, both GNPTAB and S1P are large proteins (>150 kDa) and very hard to clone. In addition, functional GNPT is a hexamer of over 340 kDa. On the other hand, much work also needs to be done from the TMEM251 side. For example, we need first to determine the membrane topology of TMEM251.

In conclusion, establishing the interaction basis between TMEM251 and S1P/GNPTAB will take a lot of effort and is well beyond the scope of the current manuscript. We thank this reviewer for this insightful suggestion of follow-up studies.

Overall, the reported findings are quite novel and highly significant on their own and warrant publication in Nature Communications.

We thank this reviewer for the time and effort in reviewing our manuscript, and we appreciate the very positive feedback and suggestions.

References:

- 1 Lenk, G. M. *et al.* CRISPR knockout screen implicates three genes in lysosome function. *Sci Rep* **9**, 9609, doi:10.1038/s41598-019-45939-w (2019).
- 2 Kikuchi, O. *et al.* Novel EGFR-targeted strategy with hybrid peptide against oesophageal squamous cell carcinoma. *Sci Rep* **6**, 22452, doi:10.1038/srep22452 (2016).
- 3 Zhang, F. *et al.* Quantification of epidermal growth factor receptor expression level and binding kinetics on cell surfaces by surface plasmon resonance imaging. *Anal Chem* **87**, 9960-9965, doi:10.1021/acs.analchem.5b02572 (2015).
- 4 Marschner, K., Kollmann, K., Schweizer, M., Bräulke, T. & Pohl, S. A key enzyme in the biogenesis of lysosomes is a protease that regulates cholesterol metabolism. *Science* **333**, 87-90, doi:10.1126/science.1205677 (2011).
- 5 van Meel, E., Qian, Y. & Kornfeld, S. A. Mislocalization of phosphotransferase as a cause of mucopolidosis III alpha. *Proc Natl Acad Sci U S A* **111**, 3532-3537, doi:10.1073/pnas.1401417111 (2014).
- 6 Ain, N. U. *et al.* Biallelic TMEM251 variants in patients with severe skeletal dysplasia and extreme short stature. *Hum Mutat* **42**, 89-101, doi:10.1002/humu.24139 (2021).

REVIEWERS' COMMENTS

Reviewer #3 (Remarks to the Author):

The authors have answered the points raised in the review satisfactory and the paper is now acceptable for publication in Nature Communications.